# Secretion and endocytosis in subapical cells support hyphal tip growth in the fungus *Trichoderma reesei*

Martin Schuster [1,3], Sreedhar Kilaru [1,3], Han A. B. Wösten [2] & Gero Steinberg [1,2] ✉

Filamentous fungi colonise substrates by invasive growth of multi-cellular hyphae. It is commonly accepted that hyphae expand by tip growth that is restricted to the first apical cell, where turgor pressure, exocytosis and endocytosis cooperate to expand the apex. Here we show that, contrary to expectations, subapical cells play important roles in hyphal growth in the industrial enzyme-producing fungus *Trichoderma reesei*. We find that the second and third cells are crucial for hyphal extension, which correlates with tip-ward cytoplasmic streaming, and the fourth-to-sixth cells support rapid growth rates. Live cell imaging reveals exocytotic and endocytic activity in both apical and subapical cells, associated with microtubule-based bi-directional transport of secretory vesicles and early endosomes across septa. Moreover, visualisation of 1,3-β-glucan synthase in subapical cells reveals that these compartments deliver cell wall-forming enzymes to the apical growth region. Thus, subapical cells are active in exocytosis and endocytosis, and deliver growth supplies and enzymes to the expanding hyphal apex.

Filamentous fungi are a major clade of eukaryotic microorganisms. They are beneficial as decomposers of organic matter[1], in mycorrhizal symbiosis with plants[2] and as industrial enzyme and small molecule producers[3]. As pathogens, filamentous fungi threaten our ecosystem, public health and food security[4,5]. A hallmark of filamentous fungi is the formation of multicellular hyphae. These structures consist of a string of cells, separated by septa that contain pores that allow cell-to-cell communication within the hypha[6]. It is widely accepted that hyphal growth is based on the tip cell (= 1st cell), which secretes and extends at its apex, thereby enabling invasive growth into the substrate or tissue[7–10]. A key process in hyphal growth is polar exocytosis of post-Golgi vesicles. These secretory vesicles (SVs) are formed at the apical Golgi compartment within the 1st cell[10] and are delivered along the cytoskeleton to the apical growth region[11–13], where they provide lipids, cell wall precursors and proteins to the growing tip[14,15]. Apical exocytosis is complemented by the endocytic uptake and recycling of lipids and proteins, which also

happens at the apex of the 1st cell[16,17], thereby focussing the growth region at the hyphal tip[18,19].

While the importance of the tip cell in hyphal growth is well-established, the role of subapical cells is less clear. Several reports suggest that hyphal tip growth does not require subapical cells[20–23]. However, an early study by Trinci reported that 7–13 subapical cells are involved in tip expansion[24]. Trinci suggested that this is due to pressure generation in these cells, which could result in tip-directed bulk flow of cytoplasm through septa[6,25–27] to extend the hyphal apex[14]. This anterograde cytoplasmic streaming (CS) can be restricted by sealing the septal apertures by Woronin bodies (WBs)[27,28]. It is also possible that subapical cells support tip growth by providing growth supplies, such as membranes and enzymes, to the expanding tip. Indeed, mitochondria, vacuoles or nuclei are able to cross septa[29], which is thought to be driven by CS[6,25]. However, in growing hyphae, growth supplies, such as cell-wall synthases, are transported in SVs and early endosomes (EEs), which deliver their content along the fibres of the

[1]Department of Biosciences, University of Exeter, Exeter, United Kingdom. [2]Microbiology, Department of Biology, University of Utrecht, Utrecht, The Netherlands. [3]These authors contributed equally: Martin Schuster, Sreedhar Kilaru. ✉e-mail: G.Steinberg@exeter.ac.uk

cytoskeleton to the apical growth region[15,30,31]. No evidence for such directed transport across septa was reported so far.

In this study, we use the ascomycete *Trichoderma reesei* to elucidate the role of subapical cells in polarised fungal growth. *T. reesei* decomposes plant debris and serves in industrial production of plant cell wall-degrading enzymes, such as cellulases[32,33]. We report here that the first and second subapical cell (hereafter named 2nd and 3rd cell) are crucial for the expansion of the tip cell, which appears to involve turgor pressure-driven CS. The subapical 4th– 6th cell increases the hyphal extension rate, which is likely to be due to tip-directed and microtubule-dependent transport of SVs and EEs across septa. Indeed, observation of the vesicle-bound v-SNARE synaptobrevin and endocytic actin patches demonstrate that secretion and endocytosis occur in subapical cells. Fluorescent recovery after photobleaching (FRAP)-experiments and photo-activation of fluorescent versions of 1,3-β-glucan synthase demonstrates that this cell wall synthase is delivered in EEs and SVs from subapical cells to the hyphal apex. Thus, subapical cells and the tip cell form a functional continuum to enable rapid growth in *T. reesei*.

## Results

### Subapical cells support hyphal tip growth in *T. reesei*

We began our analysis by laser dissecting subapical cells and measuring hyphal tip growth. For technical reasons, the analysis was limited to the apical 1st cell and the following 5 subapical cells (2nd- 6th). Moreover, unless otherwise stated, all experiments were done in growing hyphae. Individual hyphal cells were identified by labelling the plasma membrane with a fluorescent syntaxin, which revealed anticlinal septae, separating these compartments (Fig. 1a; strain QM6a_TrmsGSso1; see Supplementary Table 1 for all strains and their experimental usage, Supplementary Table 2 for plasmids description and Supplementary Fig. 1 for organisation of all plasmids constructed in this study). Cell measurements in this strain showed that the 1st cell is ~ 2- 3-times longer than the subapical cells (Supplementary Table 3).

Intercellular turgor-driven bulk flow of cytoplasm from subapical cells to the tip was suggested to support hyphal tip growth[24]. Electron microscopy showed that septa in *T. reesei* strain QM6a were perforated by wide septal pores (diameter: $225.2 \pm 28.6$ nm, $n = 17$) that are "guarded" by WBs of $290.3 \pm 69.7$ nm ($n = 24$) in diameter (Fig. 1b, "WB"). CS from subapical cells requires "open" septa, which are not "plugged" by WBs[26,27]. To visualise WBs in living cells, we identified a homologue of the *Neurospora crassa* WB protein Hex1[28] (Supplementary Table 4 for all accession numbers and bioinformatic sequence analysis). We fused TrHex1 to codon-optimised monomeric superfolder green-fluorescent protein (TrmsGFP[34]) and co-expressed the fusion protein with codon-optimised red-fluorescent mCherry (TrmCherry[34]), fused to the syntaxin homologue TrSso1[35,36]. This enabled us to distinguish between "open" septa that were guarded by several WBs and "closed" septa, where WBs concentrated in the middle of the septum (Fig. 1c and Supplementary Movie 1). Quantitative analysis of the 1st- 5th septa in *T. reesei* hyphae revealed that most septal pores are open (~ 85%; Fig. 1d). Thus, subapical cells could provide cytoplasm to the growing 1st cell.

Laser dissection of individual hyphal compartments is an accurate method for testing the contribution of subapical cells in hyphal tip expansion[22]. We grew hyphae of strain QM6a_TrmsGSso1 on potato dextrose agar surfaces and laser-injured the apical cell (Supplementary Movie 2). In ~ 53.3% of all cases ($n = 15$ hyphae), the 2nd cell formed a new growing apical cell (Supplementary Fig. 2). Next, we injured the subapical 2nd- 6th cell, followed by monitoring tip expansion at 45 min after wounding (Fig. 1e, f). Intact hyphae grew at $191.3 \pm 96.1$ μm h$^{-1}$ ($n = 180$; range: 12.0– 478.1 μm h$^{-1}$), whereas hyphal growth was almost abolished when the 2nd cell was injured and ~ 25% of the hyphae still ceased growth when the 3rd cell was injured (Fig. 1g). The extension rate of the few hyphae that were still growing was reduced by 84.9%

and 71.5%, respectively (Fig. 1h). Hyphal growth continued when the 4th– 6th cell was destroyed (Fig. 1g), yet their extension rate was significantly reduced by 59.7– 22.0% (Fig. 1h).

We noticed that sealing of septa is an inefficient process, with only ~ 80% of all 1st septa closing when the 2nd cell was injured (Supplementary Fig. 3a). Even when septa were sealed successfully, some leakage of cytoplasm from the un-wounded cell was found (Fig. 1i; arrow indicates away from the hyphal tip (= retrograde) movement of an organelle; Supplementary Movie 3). We asked if this loss accounts for impaired growth of the tip cell. To analyse this, we labelled EEs with a fluorescent version of the small GTPase Rab5 (TrmsGFP-Rab5a), a marker specific for these organelles in fungi[37,38]. Most of these EEs showed only random diffusional motility in the first cell of intact hyphae (see below), and their bulk displacement can be used as an indicator of bulk movement of cytoplasm. After wounding a 2nd cell, EEs showed bulk displacement towards the septum, which was taken as an indicator for cytoplasmic leakage from the 1st cell (Fig. 1j). We found a positive correlation of extensive cytoplasmic leakage and cell death after 45 min (Fig. 1k and Supplementary Fig. 3b; "Dying"). However, no significant difference of cytoplasm loss was found between the first cells that stopped growing (Fig. 1k, "Not growing") and the few 1st cells that continued growing after wounding the 2nd cell (Fig. 1k, "Growing"). Thus, we consider it likely that the importance of an intact 2nd cell is not due to loss of cytoplasm from the 1st cell. In summary, we conclude that the 2nd and 3rd cells are crucial for hyphal growth, whereas the 4th– 6th cell facilitate growth by increasing the rate of hyphal extension.

### Cytoplasmic streaming is restricted to the apical cell and the 2 subapical cells

Bulk flow of cytoplasm towards the hyphal tip (=anterograde direction) is thought to drive hyphal tip growth[15,24,39]. CS can be visualised by organelles that passively drift within the moving cytoplasm[25,40]. We used TrmsGFP-Rab5-labelled EEs and asked if CS occurs in growing hyphae of *T. reesei*. Most EEs showed random diffusional behaviour, while slowly drifting towards the hyphal apex (Fig. 2a). Significant anterograde CS was only found in the 1st, 2nd and 3rd cell (Fig. 2b and Supplementary Movie 4), indicating that bulk flow of cytoplasm is restricted to these 3 cells.

CS is thought to be turgor pressure-driven[39] and the decline in the 4th– 6th cell suggested that the 3rd septum obstructs retrograde turgor pressure generation, potentially by WB-independent occlusion mechanisms, reported in filamentous fungi[41,42]. Such occlusion could direct CS towards the expanding hyphal tip. We tested this idea by monitoring the diffusion of cytoplasmic TrmsGFP in strain QM6a_TrmsG_TrmChSso1. We photo-bleached either the 3rd cell or the 4th cell and analysed the fluorescent recovery, due to diffusion from the unbleached cell through the septum. We found no significant difference in anterograde or retrograde TrmsGFP diffusion (Fig. 2d, e). This suggests that the 3rd septum is equally permeable in both directions. Future studies are needed to investigate the mechanism by which the first 3 cells direct the bulk flow of cytoplasm to the hyphal tip.

### The cytoskeleton connects subapical compartments with the tip cell

Our results show that the 4th– 6th cell does not contribute significantly to anterograde CS yet are required for fast hyphal growth. We considered it possible that these cells provide growth supplies to the tip by cytoskeleton-based transport and investigate the role of the cytoskeleton in growth of *T. reesei* hyphae. We tested the importance of the cytoskeleton for growth by treating hyphae for 30 min with either the microtubule (MT) inhibitor benomyl (10 μM) or the F-actin depolymerising drug latrunculin A (10 μM). In both cases, hyphal growth was abolished (Fig. 3a). We next visualised MTs by expressing green-fluorescent α-tubulin (TrmsGFP-Tub1) in a strain containing

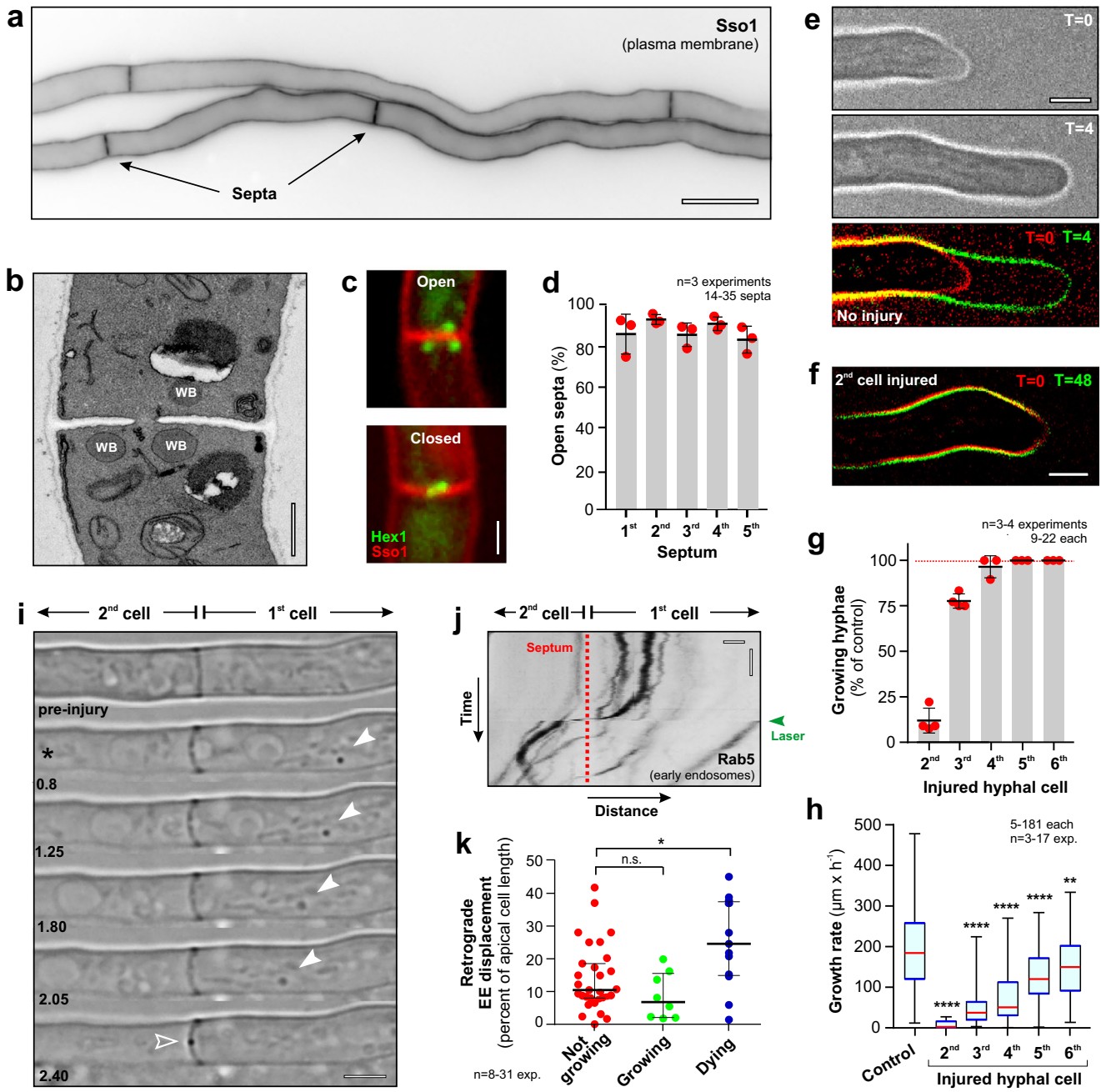

TrmCherry-Sso1. We found extended "tracks" of bundled MTs (Fig. 3b; insert shows bundling of MTs) in interphase cells. Mitotic spindles were seen occasionally in subapical and apical cells, where they co-existed with MT tracks (Supplementary Fig. 4a). This suggests that MT-based membrane trafficking continues in mitotic cells, which corresponds to previous findings in *Aspergillus nidulans*[43]. In addition, a third of all septa were connected to MTs (36.6 ± 3.3, *n* = 3 experiments, 20–22 septa each; Supplementary Fig. 4b). We next depolymerised MTs with benomyl and observed re-appearance of the tubulin polymers after transfer of the treated cells into fresh medium. We found that 25.9% of all observed septa (*n* = 27) formed an MT (Supplementary Fig. 4c). This result suggests the presence of septum-associated microtubule-organising centres. Again, such a role of septa in MT nucleation was reported in *A. nidulans*[44,45]. Thus, *T. reesei* and *A. nidulans* hyphae appear to share a similar MT organisation.

Most importantly, we found that MTs crossed septa in 50–80% of all cases analysed (1st - 5th septum; Fig. 3c, d and Supplementary

Movie 5). However, fungal MTs are dynamic structures, and with time even initially MT-free septa were crossed by new MT tracks (Fig. 3e). We also visualised the F-actin cytoskeleton, using the Lifeact peptide[46] fused to TreGFP[34]. When expressed in *T. reesei* hyphae, this reporter labelled F-actin in the hyphal tip, as well as actin patches (Fig. 3f; inset, closed arrowhead) and long actin cables, running along the length of the hyphal cell (Fig. 3f; inset, open arrowhead). Occasionally, F-actin cables crossed septa (Fig. 3g). Thus, we conclude that F-actin cables and MTs cross septa and thus could support cell-to-cell membrane trafficking.

## Subapical cells move SVs and EEs across septa

Hyphal extension depends on apical delivery of proteins, including cell wall-synthesising enzymes and membranes[15]. It is thought that newly synthesised proteins and membranes are delivered in SVs, which is complemented by endocytic recycling that involves motile EEs[31,47]. We firstly asked, if subapical cells provide SVs to the tip cell. A

**Fig. 1 | The importance of subapical cells for hyphal growth. a** Hypha of *T. reesei*, expressing TrmsGFP-Sso1. Individual cells are separated by septa. Scale bar = 10 μm. **b** Electron micrograph showing multiple WBs "guarding" the septal pore. Scale bar = 0.5 μm. **c** WBs, labelled with Hex1-TrmsGFP (green), in cells expressing the plasma membrane marker TrmCherry-Sso1 (red). WBs spread around septal pores ("Open") or locate in the centre of the septum ("Plugged"). Scale bar = 1 μm. See also Supplementary Movie 1. **d** Quantitative analysis of WB sealing of septal pores; red dots represent means of individual experiments. Note that almost all septa are open. Sample size *n* = 3 independent experiments with 14–35 septa per experiment. **e** Extension of a hypha over 4 min. False coloured overlay shows hyphal extension. Scale bar = 2.5 μm. **f** Extension of a hypha over 48 min after laser-injury of the 2nd cell. Scale bar = 2.5 μm. **g** Effect of injury of subapical cells on hyphal tip growth. Hyphae that failed to seal their apical cell after laser injury or were not growing before injury were not included. Sample size *n* = 4 independent experiments for 2nd and 3rd cell and *n* = 3 independent experiments for 4th, 5th and 6th cell with 9–22 hyphae per experiment. **h** Growth rate of intact (control) and injured hyphae of *T. reesei*. Growth was measured 45 min after laser injury. Sample size *n* = 181 hyphae for control *n* = 5 hyphae for 2nd cell, *n* = 37 hyphae for 3rd cell, *n* = 43 hyphae for 4th cell, *n* = 48 hyphae for 5th cell and *n* = 47 hyphae for 6th cell from 3–17 independent experiments. **i** Bright-field image series showing retrograde motion of cytoplasm and organelles (arrowhead) after laser injury of the

2nd cell. Note retrograde movement of cytoplasm due to leakage prior to WB-based septum sealing (open arrowhead). Scale bar = 3 μm. See also Supplementary Movie 3. **j** Contrast-inverted kymograph showing retrograde EE motility after laser injury of the second cell. Septum indicated by a red-dotted line; time of laser treatment indicated by green arrowhead. Horizontal scale bar = 1 μm; vertical scale bar = 3 sec. **k** Hyphal response to laser injury-induced cytoplasm leakage from the 1st cell. Cytoplasm leakage is given as retrograde displacement of TrmsGFP-Rab5-labelled EEs (relative to the length of the 1st cell). Growth of injured hyphae was measured after 45 min. Sample size *n* = 31(Not growing), *n* = 8 (Growing) and *n* = 13 (Dying) independent experiments. See also Supplementary Fig. 2. Results were confirmed in 3 (**a, c, e, f, i, j**) and 2 (**b**) independent experiments. Bars in (**d, g**) represent mean ± SEM; sample sizes are indicated; red dots represent independent experiments; some data sets in (**h,k**) did not pass a normality test (Shapiro-Wilk test, *P* > 0.05) and thus are given as Whiskers' plots (**h**, blue lines: 25/75 percentiles; red line: median; minimum and maximum at whiskers ends) or as individual data points, including median and 95% confidence interval (**k**); statistical comparison with control in (**h, k**) used non-parametric Mann-Whitney testing, with *= two-tailed *P*-value 0.0134 (**k**), **= two-tailed *P*-values of 0.0059 (**h**) and ****= two-tailed *P*-values < 0.0001 (**h**). Data in (**a, d–k**) were obtained from growing hyphae. All data are provided in the Source Data file.

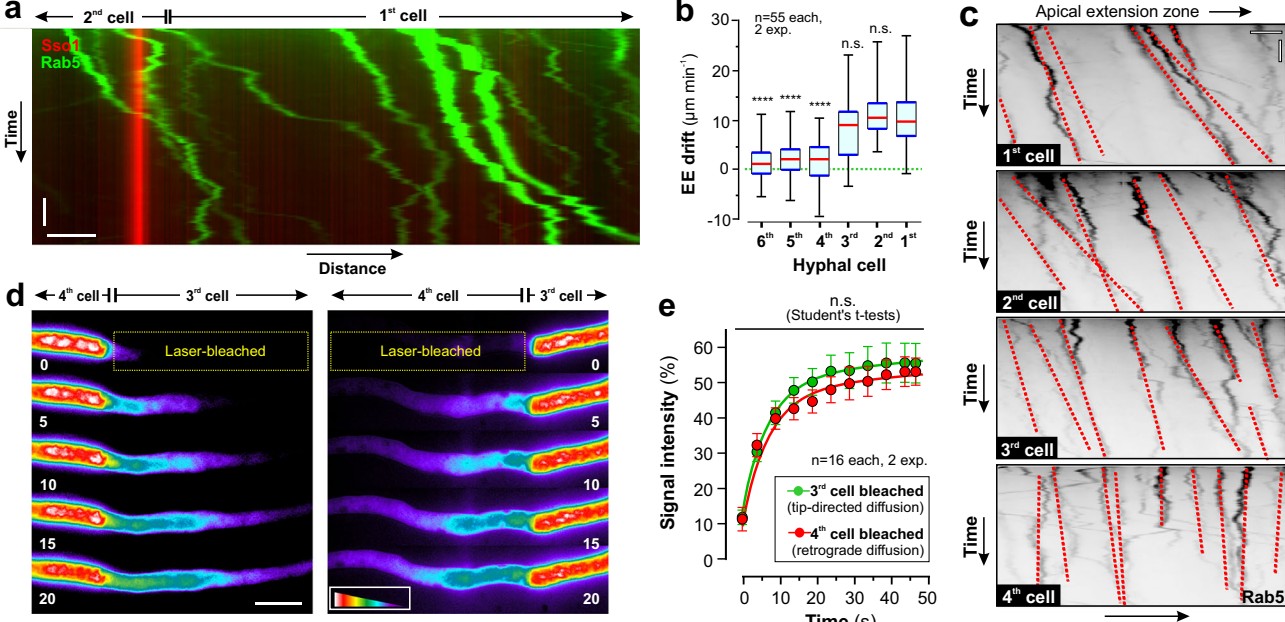

**Fig. 2 | Cytoplasmic streaming and osmotic pressure in *T. reesei*. a** Kymograph showing drift of EEs, labelled with TrmsGFP-Rab5 (green), in a growing hypha; septum is labelled with TrmCherry-Sso1 (red). Note that most organelles slowly drift towards hyphal tip. Horizontal scale bar = 2 μm; vertical scale bar 3 s. See also Supplementary Fig. 6. **b** Diffusional behaviour of EEs in the first 6 cells of hyphae. Organelles that show directed transport were excluded from the analysis. Note that tip-directed drift of EEs was only found in the 1st, 2nd and 3rd cell. Sample size *n* = 55 hyphae from 2 independent experiments. **c** Contrast-inverted Kymographs of EEs, labelled with TrmsGFP-Rab5, in the 1st- 4th cell. Red-dotted lines indicate the drift displacement of organelles. Horizontal scale bar = 2 μm; vertical scale bar = 3 s. See also Supplementary Movie 4. **d** Diffusion of cytoplasmic TrmsGFP through the 3rd septum. The 3rd or 4th cell was photo-bleached (darkened area indicated by yellow box) and diffusion of unbleached TrmsGFP into the darkened area was monitored. Time after photo-bleaching is given in seconds under each frame. Grey-scale

images are given in pseudo-colours, the intensity scale is given in the lower left corner of the right panel. Scale bar = 10 μm. See also Supplementary Movie 5. **e** Quantitative analysis of cytoplasmic GFP fluorescence at 5 μm adjacent to the septum in the photo-bleached 3rd cell (green) or 4th cell (red). Sample size *n* = 16 hyphae from 2 independent experiments. Results shown in (**a, c, d**) were confirmed in 2 independent experiments. Some data sets in (**b**) did not pass a normality test (Shapiro-Wilk test, *P* > 0.05) and thus are given as Whiskers' plots (blue lines: 25/75 percentiles; red line: median; minimum and maximum at whiskers ends), Data in (**e**) show mean ± SEM; statistical comparison with control in (**b**) used non-parametric Mann-Whitney testing, comparison of data sets in (**e**) used one-way ANOVA testing, n.s. = non-significant difference (**b, e**) and ****= two-tailed *P*-values < 0.0001 (**b**). Data in (**a–e**) were obtained from growing hyphae of *T. reesei*. All data are provided in the Source Data file.

marker for post-Golgi fungal SVs is the small GTPase Sec4[48–50]. We identified a *sec4* homologue in the genome of *T. reesei* (TrSec4; Supplementary Table 4) and fused 2x TrmsGFP to the N-terminus of the endogenous gene. Co-visualisation of the plasma membrane and fluorescent TrmsGFP2Sec4 showed that most SV are concentrated in

the 1st cell (Fig. 4a). We found rapid bi-directional movement of SVs in the 1st and subapical cells (Fig. 4b and Supplementary Fig. 5a, b). Anterograde motility of SVs was most prominent in the 1st cell (Fig. 4c), yet laser-bleaching experiments revealed extensive retrograde movement of SVs in the apical 20 − 50 μm (Supplementary

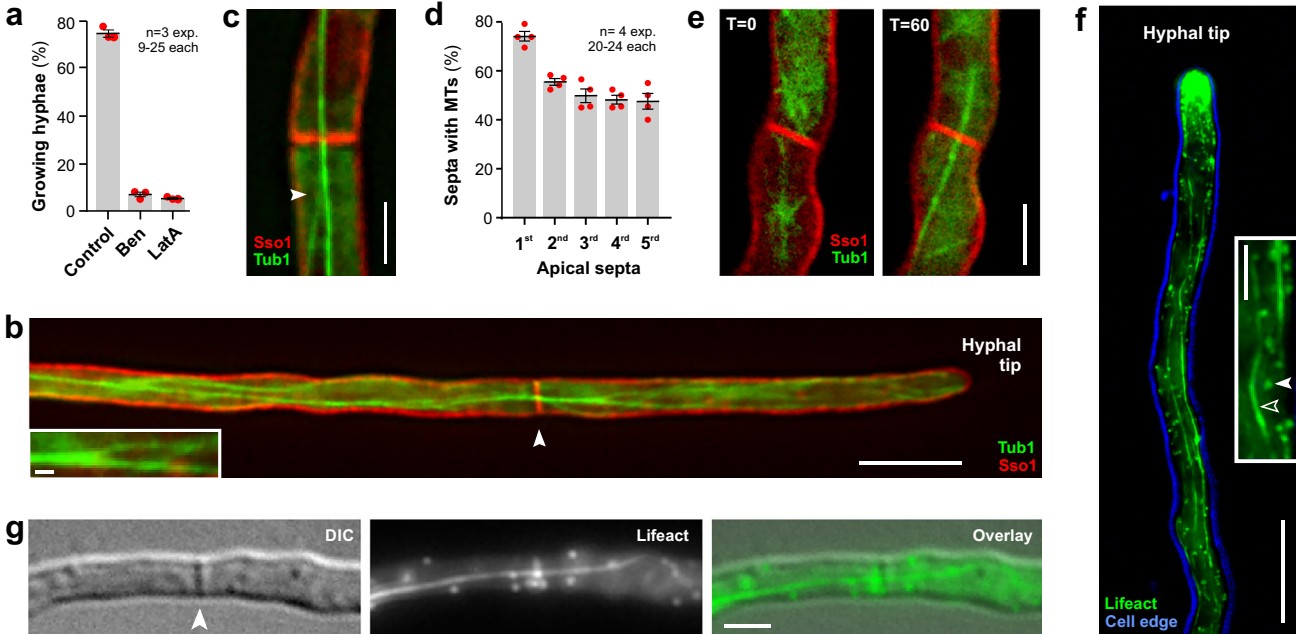

**Fig. 3 | The cytoskeleton in *T. reesei* hyphae. a** Effect of cytoskeleton inhibitors on hyphal growth. Growth was measured after 30 min treatment with the corresponding amounts of the solvent DMSO (control), 10 μM of the MT inhibitor benomyl or 10 μM of the F-actin inhibitor Latrunculin A (LatA). Sample size $n = 3$ independent experiments with 9–25 hyphae each. **b** Microtubules (MTs), labelled with TrmsGFP-Tub1 (green), in a hypha of *T. reesei*. The plasma membrane is labelled by TrmCherry-Sso1 (red); a septum is indicated by an arrowhead. Scale bars = 5 μm (overview), 0.5 μm (inset). **c** MTs that extend through a septal pore. Several MTs merge to form a bundle (arrowhead). Scale bar = 2 μm. See also Supplementary Movie 5. **d** Quantitative analysis of cell-connecting MTs in the 5 septa behind the apical 1st cell. Note that at a given moment, only ~ 50–80% of all septa are crossed by an MT track. Sample size $n = 4$ independent experiments with 20–24 hyphae each. **e** Reorganisation of MTs (green) at a septum (red) over time. A septum shows no MT running through the septal pore ($t = 0$). After 60 min ($t = 60$), a MT track crosses the same septum and now connects the 2 adjacent cells. Scale bar= 2 μm. **f** Filamentous actin (F-actin) in *T. reesei*. Lifeact-TreGFP labelled F-actin concentrates at the hyphal tip, forms small actin-patches (filled arrowhead in inset) and long actin cables (open arrowhead in inset). The cell edge is indicated in blue. Scale bars = 10 μm (overview) and 2 μm (inset). **g** F-actin cable that crosses a septum (indicated by arrowhead). Scale bar = 2 μm. Results shown in (**b**, **c**) were confirmed in 4 independent experiments, in (**e**–**g**) in 2 independent experiments. Data in (**a**, **d**) show mean ± SEM sample sizes; red dots represent independent experiments. Data in (**a**–**e**, **h**–**k**) were obtained from growing. All data are provided in the Source Data File.

Fig. 5c and Supplementary Movie 6). The significance of this retrograde transport of SVs need to be addressed in a separate study. Treatment with benomyl or latrunculin A revealed that this motility required MTs but not F-actin (Fig. 4d and Supplementary Movie 7). SVs occasionally crossed septa in both directions (Fig. 4e, f and Supplementary Movie 8), suggesting that apical and subapical cells exchange newly synthesised proteins and lipids. Thus, subapical cells could participate in growth by providing growth supplies to the hyphal tip.

Next, we investigate if EE trafficking occurs between hyphal cells. Consistent with a role in hyphal growth[18,31,47], fluorescent EEs concentrated in the 1st cell (Fig. 5a). In contrast to SVs, most EEs showed diffusional and random motility, while a small number was rapidly moving bi-directionally and over long distances (Supplementary Fig. 6; arrowheads). This behaviour was also found in subapical cells (Fig. 5b, 4th cell shown as example; Supplementary Movie 9), although the frequency of EE motility was reduced (Fig. 5c). Pharmacological experiments revealed that directed EE motility was MT dependent (Fig. 5d and Supplementary Movie 10), which is consistent with findings in other fungi[31,51,52]. Rapidly moving EEs were crossing septa (Fig. 5e and Supplementary Movie 11). This cell-to-cell motility was found to be bi-directional, yet significantly more EEs moved in anterograde direction (Supplementary Fig. 7). Intercellular trafficking of EEs was found to be most prominent at the 1st septum (Fig. 5f). These results support the idea that subapical cells could support hyphal tip growth by delivery of recycled proteins and membranes in EE to the apical growth region.

## Endocytosis is prominent in subapical cells

The anterograde net flow of EEs from subapical cells to the tip implies that endocytosis occurs in subapical cells. Such activity was never reported, as endocytosis in filamentous fungi is thought to occur at tip of the apical 1st cell[18,19]. To address this, we performed pulse-chase experiments with the endocytosis reporter FM4-64. The dye inserts into the plasma membrane, travels through the endocytic pathway of the cell, and finally ends up in the membrane of the fungal vacuole[47,53]. When applied to *T. reesei* hyphae, FM4-64 reached DCFA-stained vacuoles in the tip-cell (Fig. 6a, 1st cell). However, the dye also ended up in vacuolar membranes in all 5 subapical cells investigated (Fig. 6a; 2nd– 6th cell). These results provided the first indication that endocytosis is not restricted to the apical cell. We next set out to investigate early steps in endocytosis in *T. reesei* by observing actin patches, which are involved in formation of endocytic vesicles at the plasma membrane[54–56]. Consistent with the notion that endocytosis occurs in subapical cells, we found actin patches in the 2nd cell (Fig. 6b). They appeared and disappeared at the plasma membrane, suggesting that endocytic vesicles are formed (Fig. 6c). This dynamic behaviour is best visible in kymographs, where a stationary signal appears during formation, followed by a switch to random diffusional and/or directed motility upon release of the endocytic vesicle into the cytoplasm, a behaviour that was first described in the yeast *Saccharomyces cerevisiae*[54,57] (Fig. 6d; left panel represents a side view, right panel was derived from a top view). We found dynamic actin patches in the 1st and all subapical cells (Fig. 6e and Supplementary Movie 12). Quantitative analysis of actin patch numbers revealed that most endocytic activity

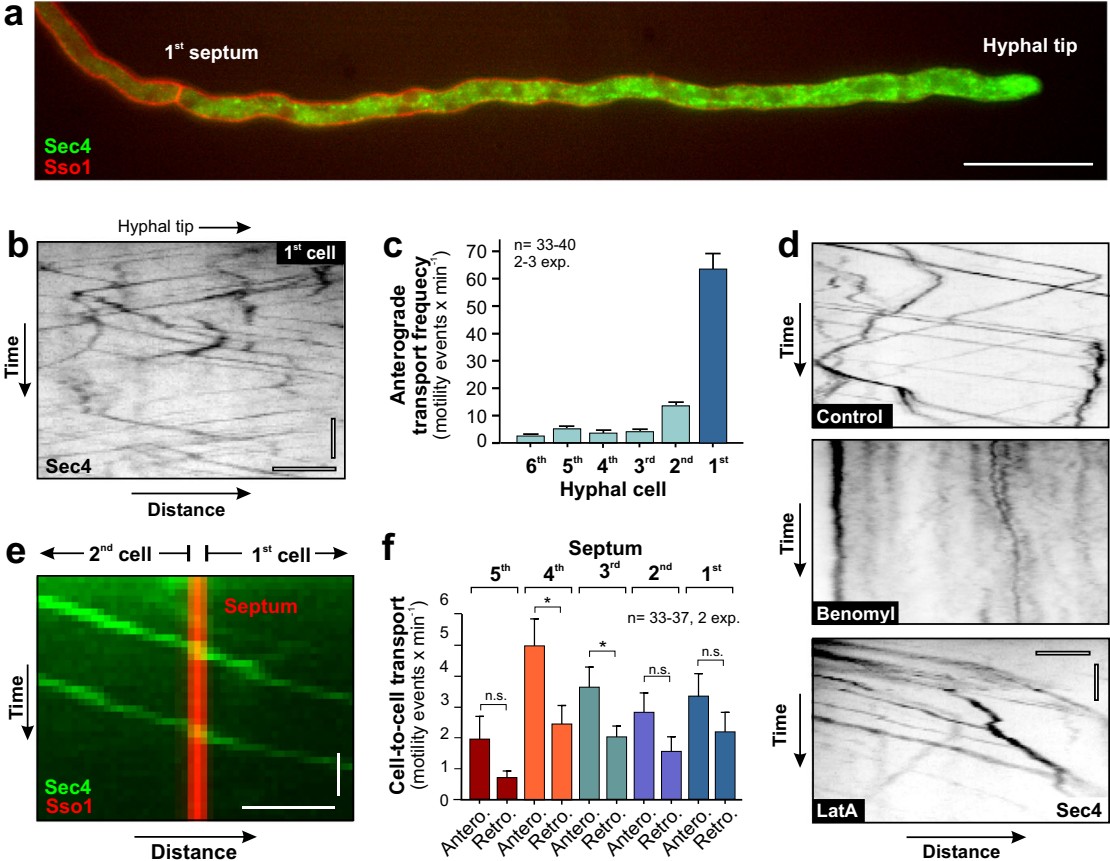

**Fig. 4 | SV transport towards the growing hyphal tip in *T. reesei*. a** SVs, labelled with TrmsGFP₂-Sec4, in the 1st hyphal cell in *T. reesei*. Note that the endogenous *sec4* gene of *sec4* was fused to the fluorescent tag; consequently, all SVs are fluorescently labelled. The plasma membrane is labelled by TrmCherry-Sso1 (red). Scale bar = 20 μm. **b** Bidirectional motility of SVs within the 1st cell. Note that the contrast-inverted kymograph was taken at a rear region of the cell as high density of SVs in the apical region of the 1st cell challenge visualisation of SV motility. Horizontal scale bar = 2 μm; vertical scale bar = 3 s. **c** Quantitative analysis of anterograde motility of SVs in the tip cell and 5 subapical cells. Numbers were estimated in un-bleached hyphae; the region of measurements in subapical cells were located 10 μm behind the septum; numbers in the 1st cells were determined in regions located 10 μm behind the tip. Sample size *n* = 40 for 1st and 2nd cell, *n* = 33 for 3rd cell, *n* = 37 for 4th cell and *n* = 35 for 5th and 6th cell from 2–3 independent experiments. **d** Contrast-inverted kymographs showing the effect of inhibitors of the cytoskeleton motility of SVs, labelled with TrmsGFP₂-Sec4. Motility ceases when microtubules are depolymerised (benomyl), but continued in the absence of F-actin (LatA). Cells were treated 30 min with 10 μM of benomyl, 10 μM latrunculin A or the corresponding amounts of the solvent DMSO (control). Horizontal scale bar = 3 μm; vertical scale bar = 3 s. See also Supplementary Movie 7. **e** Kymograph showing motility of SVs, labelled with TrmsGFP₂-Sec4 (green), across the 1st septum, labelled with TrmCherry-Sso1 (red). Horizontal scale bar = 1 μm; vertical scale bar = 1 s. See also Supplementary Movie 8. **f** Quantitative analysis of anterograde and retrograde motility of SVs across septa in hyphae. Only directed motility was included, passive drifting of SVs through the septum was not considered. Sample size *n* = 33 for 1st septum, *n* = 34 for 2nd and 5th septum, *n* = 37 for 3rd septum and *n* = 35 for 4th septum from 2 independent experiments. Results were confirmed in 3 (**a**, **b**) and 2 (**d**, **e**) independent experiments. Bars in (**c**, **f**) represent mean ± SEM; sample sizes are indicated. Statistical comparison with in (**f**) used Student's *t* testing testing; n.s. = non-significant difference, *= two-tailed *P*-values 0.0444 (3rd) and *P*-values 0.0181 (4th). All data were obtained from growing hyphae and are provided in the Source Data File.

occurs at the apical dome of the 1st cell (Fig. 6f; 1st cell, tip). This number declined in subapical regions, yet significant endocytic activity was found in all subapical cells (Fig. 6f; 2nd–6th cell; Table 1). These results are consistent with the hypothesis that endocytic recycling in subapical cells supports hyphal tip growth in *T. reesei*.

**Secretion occurs in subapical cells**

Endocytosis in subapical cells requires subapical exocytosis to maintain a balance of proteins and membranes in the plasma membrane. Membrane lipids are synthesised in the endoplasmic reticulum (ER)[58], and protein secretion involves the Golgi apparatus[59]. We labelled both compartments in strain QM6a with specific fluorescent reporter proteins (ER: a codon-optimised enhanced GFP, N-terminally fused to a signal sequence from rabbit calreticulin and C-terminally fused to the ER retention signal HDEL at its C-terminal end; Golgi: a fluorescent homologue of the α−1,2-mannosyltransferase Ktr1[60]; Supplementary Fig. 1h, i). We found that apical cells have the highest density of ER

network and Golgi vesicles, yet both compartments are also prominent in subapical cells (Fig. 7a, b, and Supplementary Fig. 8). Thus, we conclude that subapical cells of the hypha have the biosynthetic compartments to synthesise membranes and proteins.

Next, we attempted to visualise secretion events at the plasma membrane by observing a vesicle-associated synaptobrevin homologue Snc1[35] (Supplementary Table 4), which mediates exocytosis at the plasma membrane[61]. A similar approach revealed active secretion at the hyphal tip of *A. nidulans*[17]. An in-locus fusion of TrmsGFP and *snc1* revealed that most of the v-SNARE was concentrated in the apical region of the 1st cell (Fig. 7c). Co-localisation of a red-fluorescent TrmCherrySnc1 with TrmsGFP₂-Sec4 showed that the synaptobrevin travels in SVs (Fig. 7d). We also found TrmCherrySnc1 that moved independently of TrmsGFP₂-Sec4 (Fig. 7d, arrowheads; Supplementary Movie 13), suggesting that the v-SNARE also localises to endocytic recycling vesicles. We next set out to observe fusion of vesicles with the plasma membrane in subapical cells, which is expected to result in

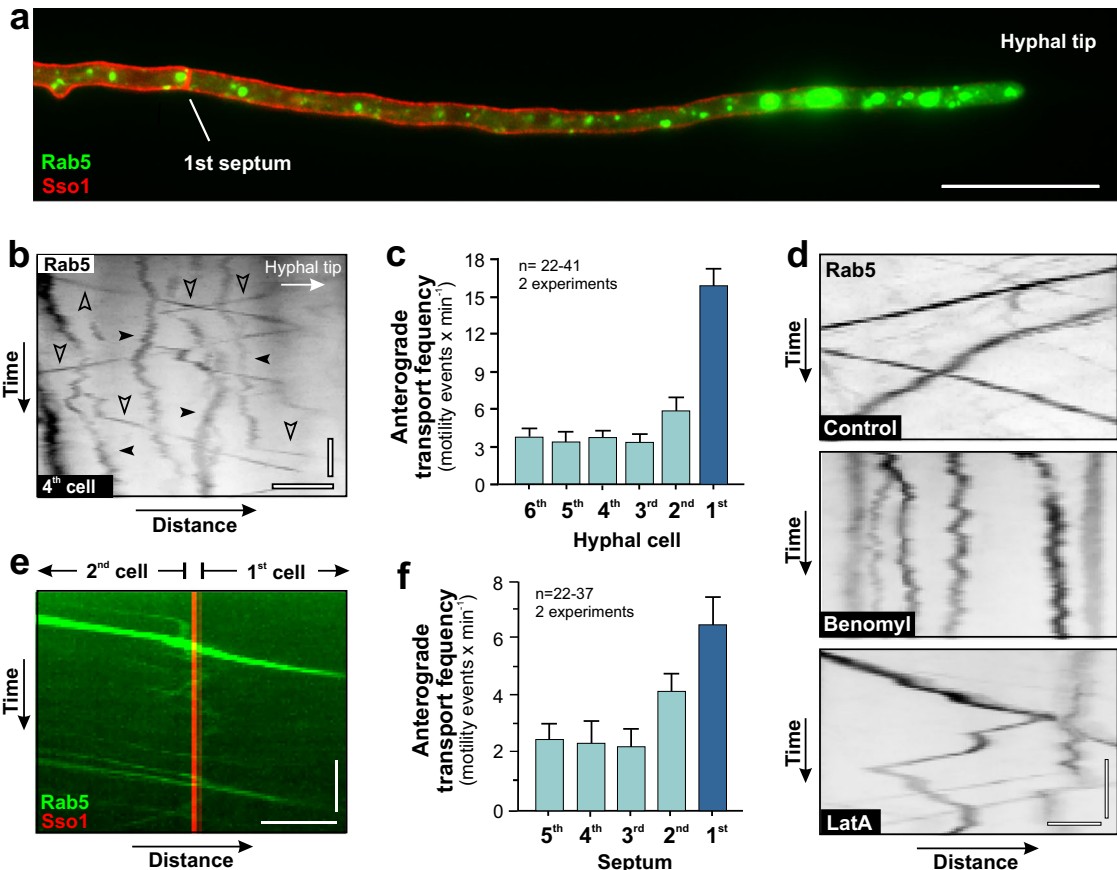

**Fig. 5 | EE transport towards the growing hyphal tip in *T. reesei*. a** EEs, labelled with TrmsGFP-Rab5, in a 1st cell. Scale bar = 20 μm; plasma membrane is labelled by TrmCherry-Sso1 (red). **b** Motility of EEs (TrmsGFP-Rab5) in a 4th cell; diffusional motility indicated by filled arrowhead, directed motility indicated by open arrowhead. Horizontal scale bar = 2 μm; vertical scale bar = 3 s. See also Supplementary Movie 9. **c** Quantitative analysis of anterograde motility of EEs in the 1st cell (dark blue) and following subapical cells; measurements in subapical cells were done 10 μm behind the septum; numbers in the 1st cells were determined 25 μm behind the apex. Sample size *n* = 41 for 1st cell, *n* = 22 for 2nd cell, *n* = 36 for 3rd cell *n* = 35 for 4th and 5th cell and *n* = 37 for 6th cell from 2 independent experiments. **d** Contrast-inverted kymographs showing the effect of microtubule inhibitor benomyl or the F-actin inhibitor latrunculin A (LatA) on EE motility. Cells were treated for 30 min

with 10 μM inhibitor or corresponding amounts of the solvent DMSO (control). Horizontal scale bar = 1.5 μm; vertical scale bar = 3 s. See also Supplementary Movie 10. **e** Kymograph showing motility of EEs (TrmsGFP-Rab5; green), across the 1st septum, labelled with the plasma membrane marker TrmCherry-Sso1 (red). Horizontal scale bar = 2 μm; vertical scale bar = 3 s. See also Supplementary Movie 11. **f** Quantitative analysis of directed anterograde motility of EEs across septa. Sample size *n* = 22 for 1st septum, *n* = 35 for 2nd and 3rd septum, *n* = 34 for 4th septum and *n* = 37 for 5th septum from 2 independent experiments. Results shown in (**a, b, d, e**) were confirmed in 2 independent experiments. Bars in (**c, f**) represent mean ± SEM; sample sizes are indicated. Data in (**a–f**) were obtained from growing hyphae. All data are provided in the Source Data file.

dispersal of the v-SNARE signal. Co-visualisation of TrmCherrySso1 and TrmsGFPSnc1 in the 2nd cell revealed that vesicles are rapidly travelling along the cell and pause for a short time underneath the plasma membrane, before the signal moves into the plasma membrane and disappears (Fig. 7e; open arrowhead; Supplementary Movie 14). This dynamic behaviour was confirmed by top-views in total internal reflection fluorescence microscopy (TIRFM), which showed that signals appeared at the plasma membrane and remained stationary for 2.9 ± 0.4 s (n = 59), before they disappeared (Fig. 7f, g). This behaviour is consistent with the microtubule-based delivery of a vesicle, tethering underneath the plasma membrane, and subsequent exocytosis, which disperses the formed v-SNARE/t-SNARE complex[61]. Finally, we labelled the endogenous copy of a *T. reesei* homologue of the exocyst subunit Exo70 (Supplementary Table 4 for bioinformatic analysis) with codon-optimised eGFP. Exo70p in *S. cerevisiae* locates at the plasma membrane, where it marks sites of exocytosis[49,62]. We found TreGFP-TrExo70 signals in the plasma membrane of 2nd cells in *T. reesei* hyphae (Fig. 7h), which is consistent with the notion of exocytosis in subapical cells. Thus, we conclude that secretion replenishes the plasma membrane of subapical cells with lipids and proteins.

## Subapical cells provide 1,3-β-glucan synthase to the apical hyphal growth zone

Our results suggest that growth supplies, such as cell wall synthases, are provided by subapical cells. 1,3-β-glucan synthase is such a cell wall synthesising enzyme, which is crucial for hyphal growth[63]. We identified a putative 1,3-β-glucan synthase gene in the genome of *T. reesei* strain QM6a (TrGcs1; Fig. 8a and Supplementary Table 4). We fused the endogenous *gcs1* gene to a double tag of codon-optimised mCherry and expressed the fluorescent glucan synthase in a strain that contained green-fluorescent plasma membrane (TrmsGFP-Sso1). Co-observation of both markers revealed that the fluorescent cell wall synthase concentrates at the hyphal tip (Fig. 8b). We next set out to visualise anterograde transport of the cell-wall synthase in SVs in a strain that expressed TrmCherry₂-Gcs1 and TrmsGFP₂-Sec4. We found that only ~ 39.6% of the red-fluorescent signals co-travelled with the green-fluorescent signals (*n* = 3 experiments, 11–28 signals each; Fig. 8c, Supplementary Fig. 9a and Supplementary Movie 15). This suggested that 1,3-β-glucan synthase is travelling in SVs and endocytic membranes. Endocytic recycling of 1,3-β-glucan synthase was shown in *A. nidulans*[64]. We therefore tested if TrmCherry₂-Gcs1 is transported in

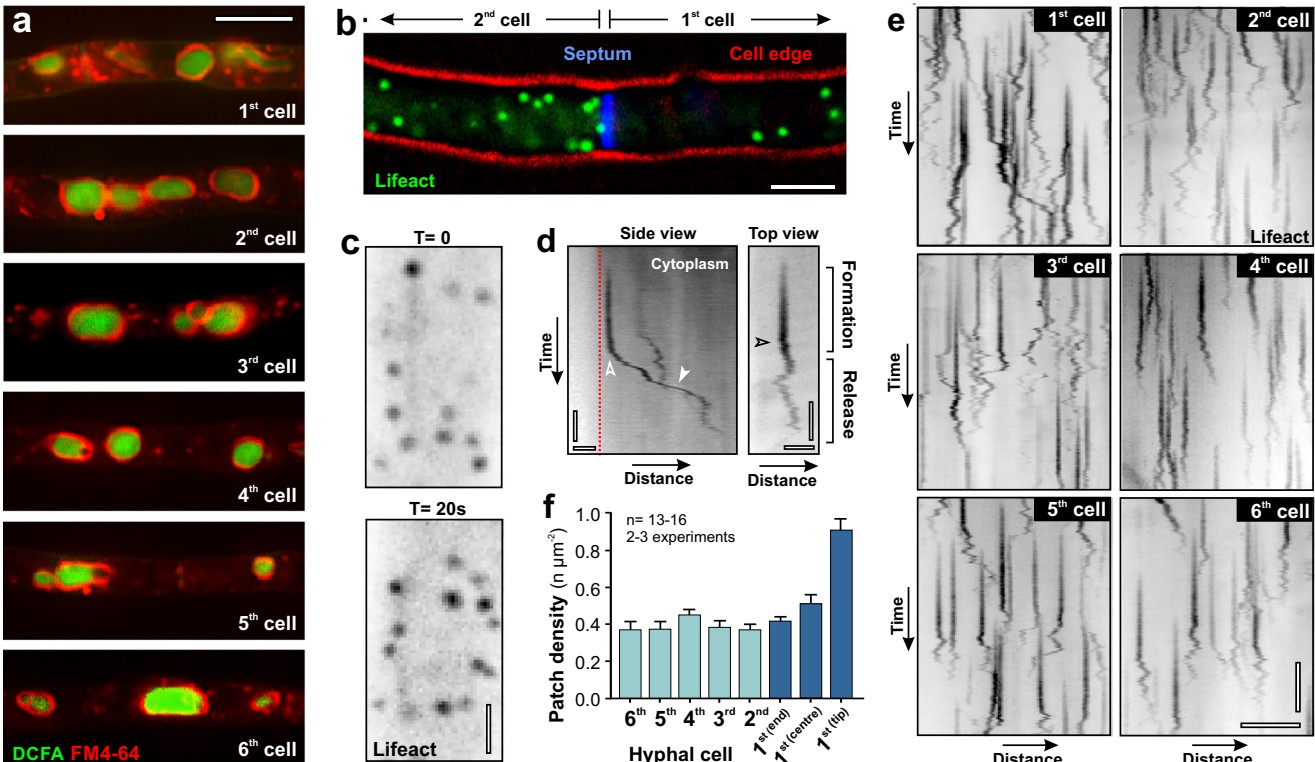

**Fig. 6 | Endocytosis in subapical cells. a** FM4-64 staining of hyphal cells, co-stained with the vacuole marker DCFDA. Note that the dye reaches vacuolar membranes due to endocytic uptake. Scale bar = 5 μm. **b** Endocytic actin patches, labelled with Lifeact-TreGFP, at the subapical region of the 1st cell and in the 2nd cell. The edge of the cell is shown in red; the 1st septum is stained using Calcofluor White (blue). Scale bar = 2 μm. **c** Contrast-inverted image of actin patches in the 2nd cell. Time in seconds between images is shown in the upper right corner. Note that patch dynamics indicate active endocytosis. Scale bar = 1 μm. **d** Contrast-inverted kymographs showing the dynamic behaviour of an actin patch at the plasma membrane (left panel: side view; right panel: top view). The patch appears at the plasma membrane and is stationary during endocytic vesicle (Formation). Shortly after appearance, the actin patch shows diffusional motility (transition indicated by

open arrowheads) and occasionally undergoes rapid directed transport (closed arrowhead). Shortly after this, the actin patch disappeared. Horizontal scale bars = 1 μm; vertical scale bars = 3 s. **e** Contrast-inverted kymographs showing the dynamic behaviour of an actin patch at the plasma membrane in the apical and all subapical cells. Data were taken 20 μm behind the tip or septa. Horizontal scale bar = 3 μm; vertical scale bar = 5 s. See also Supplementary Movie 12. **f** Distribution of actin patches, labelled with Lifeact-TreGFP, in hyphae of *T. reesei*. Sample size n = 13- cells for $1^{st(tip)}$, $1^{st(centre)}$, $2^{nd}$, $3^{rd}$, $4^{th}$, $5^{th}$, $6^{th}$ and Sample size $n = 16$ for $1^{st(end)}$ from 2–3 independent experiments. Results shown in (**a**–**e**) were confirmed in 2 independent experiments. Data in (**f**) show mean ± SEM. Data in (**a**) were obtained from growing hyphae. All data are provided in the Source Data file.

### Table 1 | Actin patch number and endocytic activity in *T. reesei*

| Name | Length[a] (μm) | Width[b] (μm) | Cellular surface area[c] (μm³) | Endocytic activity[d] (actin patches μm⁻²) | Actin-patches per cell | Relative activity in cell[f] (%) | Relative activity in 1 um²,[f] (%) |
|---|---|---|---|---|---|---|---|
| 1st cell | 188.8 ± 40.6 | 2.42 ± 0.45 | 1435.38 | 0.61 ± 0.28[e] | 875.45 | 100.00 | 100.00 |
| 2nd cell | 74.1 ± 17.7 | 2.53 ± 0.49 | 588.96 | 0.37 ± 0.11 | 218.30 | 25.02 | 56.27 |
| 3rd cell | 71.4 ± 19.2 | 2.55 ± 0.37 | 571.99 | 0.38 ± 0.13 | 218.85 | 25.08 | 45.71 |
| 4th cell | 76.3 ± 23.5 | 2.46 ± 0.28 | 589.67 | 0.45 ± 0.11 | 265.08 | 30.38 | 40.77 |
| 5th cell | 88.2 ± 20.7 | 2.50 ± 0.24 | 692.72 | 0.37 ± 0.16 | 257.80 | 29.55 | 42.08 |
| 6th cell | 84.0 ± 24.1 | 2.60 ± 0.35 | 686.12 | 0.37 ± 0.11 | 254.18 | 29.14 | 49.45 |

[a] $n = 60$ cells each.
[b] $n = 20$–27 cells each.
[c] Estimated outer surface area of a tube, calculated at https://onlinevonversion.com/object_surfacearea_tube.htm.
[d] Given as number of mean ± STDEV actin patch number per 1 μm² ($n = 13$–16).
[e] Average of the number in the apical 5 μm (0.87 ± 0.20, $n = 13$), the middle region (0.54 ± 0.18, $n = 15$) and 5–10 μm before the septum (0.42 ± 0.10, $n = 13$).
[f] Activity in the entire cell; for the tip cell, the average of 3 regions was used.
[g] Relative endocytic activity compared to the that in hyphal apex.

TrmsGFP-Rab5-carrying EEs. Indeed, many red-fluorescent cell wall synthase co-travelled with the green-fluorescent EE marker (Fig. 8d, Supplementary Fig. 9b and Supplementary Movie 16) and even crossed septa (Supplementary Movie 17). This observation further supports the notion that endocytic membrane recycling and EEs contribute to hyphal tip growth.

We next tested if subapical cells provide 1,3-β-glucan synthase to the apical hyphal expansion zone. We firstly performed a series of FRAP experiments to investigate the dynamic behaviour of fluorescent TrmCherry₂-Gcs1. We found TrmCherry₂-Gcs1 in the plasma membrane of subapical cells (Fig. 8e). These signals returned into the plasma membrane of these cells within 20 min after photo-bleaching

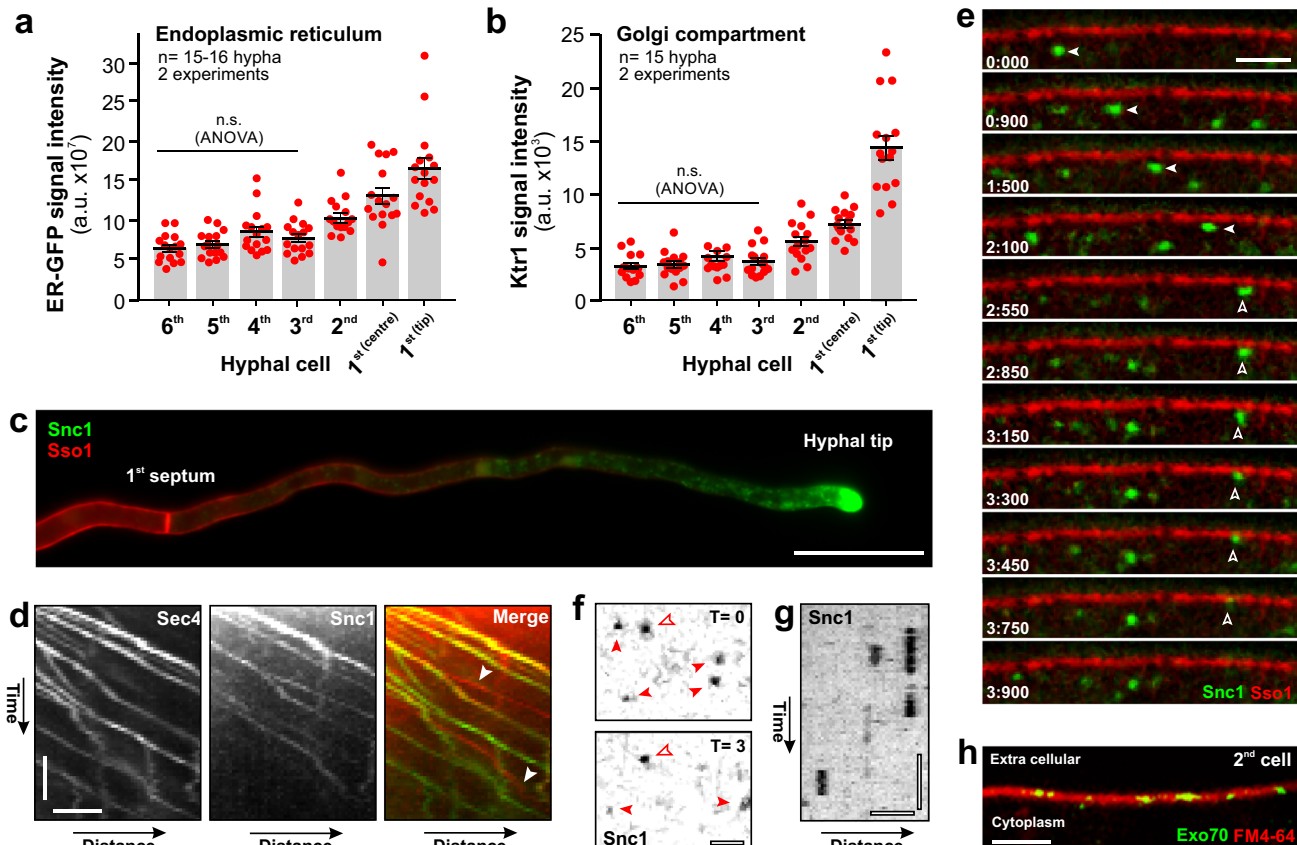

**Fig. 7 | Distribution of ER and Golgi and vesicle fusion at plasma membrane of subapical cells. a** Distribution of ER, labelled with the fluorescent marker ER-GFP (TreGFP-HDEL), in growing *T. reesei* hyphae. The average fluorescent signal intensity was measured in sum-projection images in the middle region of the cells or 10 μm behind the hyphal tip (1st(tip)). See also Supplementary Fig. 8a. Sample size *n* = 15–18 hyphae from 2 independent experiments. **b** Distribution of Golgi vesicles, labelled with TrKrt1-TreGFP, in growing *T. reesei* hyphae. The average fluorescent signal intensity was measured in sum-projection images in the middle region of the cells or 10 μm behind the hyphal tip (1st(tip)). See also Supplementary Fig. 8b. Sample size *n* = 15 hyphae from 2 independent experiments. **c** Distribution of the fluorescent synaptobrevin-like v-SNARE TrmsGFP-Snc1 in the 1st cell of a hypha. Note that the endogenous *snc1* gene was fused to codon-optimised monomeric super-folder GFP. Scale bar = 20 μm. **d** Kymograph showing anterograde co-motility of TrmsCherry-Snc1 (red in merged image) and the SV marker TrmsGFP$_2$-Sec4 (green in merged image) in a 1st cell. In the merged image, co-localisation results in yellow lines. Note that some TrmsCherry-Snc1 signals did not travel on SVs (arrowheads). Horizontal scale bar = 2 μm; vertical scale bar = 2 s. See also Supplementary Movie 13. **e** Image series showing delivery and plasma membrane fusion of a TrmsGFP-Snc1-bound vesicle in a subapical 2nd cell. Time in seconds:milliseconds is given in the upper left corner of each panel. The vesicles rapidly move along the periphery of the subapical cell (*T* = 0:000 to 3:450); after tethering underneath the plasma membrane (T = 3:450 to 4:050) the signal disappears *T* = 4:200 to 4:700). Scale bar = 2 μm. **f** Top view of TrmsGFP-Snc1 at the plasma membrane in a subapical 2nd cell. Images were taken using TIRFM; time between images is 3 s. Scale bar= 1 μm. See also Supplementary Movie 14. **g** Kymograph showing dynamic behaviour of TrmsGFP-Snc1 signals at the plasma membrane, visualised using TIRFM. Horizontal scale bar = 1 μm; vertical scale bar = 2 s. **h** Image showing the location of the exocyst subunit Exo70 (green) at the plasma membrane of a subapical 2nd cell (labelled with a short pulse of FM4-64). Note that the endogenous gene for *T. reesei* Exo70 homologue was tagged with codon-optimised enhanced GFP. Scale bar= 2 μm. Results shown in (**c**–**h**) were confirmed in 2 independent experiments. Bars in (**a**, **b**) represent mean ± SEM; sample sizes are indicated; red dots in (**a**, **b**) represent independent measurements; statistical comparison in (**a**, **b**) used one-way ANOVA testing; n.s.= non-significant difference. Data in (**a**–**h**) were obtained from growing hyphae. All data are provided in the Source Data file.

(Fig. 8f). This result is consistent with exocytosis in subapical cells. Next, we photo-bleached the entire 1st cell and monitored fluorescent recovery of 1,3-β-glucan synthase with time. To avoid the synthesis of a new enzyme, we performed these experiments in the presence of 100 μM cycloheximide, shown to prevent protein synthesis in *T. reesei*[65]. Fluorescent 1,3-β-glucan synthases from the subapical cells appeared at the hyphal apex of the photo-bleached 1st cell (Fig. 8g). Some of these signals inserted into the apical plasma membrane (Fig. 8h), where they remained stationary (Fig. 8i). This behaviour is thought to be linked to the formation of newly synthesised cell wall polymer[66], suggesting that the delivered enzymes participate in cell wall synthesis at the extending hyphal apex.

In a second set of experiments, we used a fusion of photo-activatable GFP and Gcs1 (paGFP-Gcs1) and photo-activated the fusion protein in the subapical 2nd cell. Before and directly after photo-activation, very faint fluorescence was visible at the hyphal apex (Fig. 8j). However, 30–50 min after laser-based photo-activation, fluorescent 1,3-β-glucan synthases accumulated at the tip (Fig. 8j). No such accumulation was found in control experiments, where no photo-activation occurred. This result further supports the notion that sub-apical cells provide wall-synthases to the growing hyphal tip. Finally, we stained hyphae with the apical paGFP-Gcs1 signals with FM4-64. Shortly after application of the endocytic dye, red-fluorescence co-localised with paGFP-Gcs1, suggesting that the 1,3-β-glucan synthase resides in endocytic membranes (Fig. 8k). However, some paGFP-Gcs1 did not co-localise with FM4-64 (Fig. 8k, arrowhead), which suggests that a small portion of newly synthesised cell-wall synthase is also delivered from the 2nd cell to the hyphal tip. Taken together, our experiments indicate that subapical cells support tip growth by delivering cell-wall synthases to the hyphal apex.

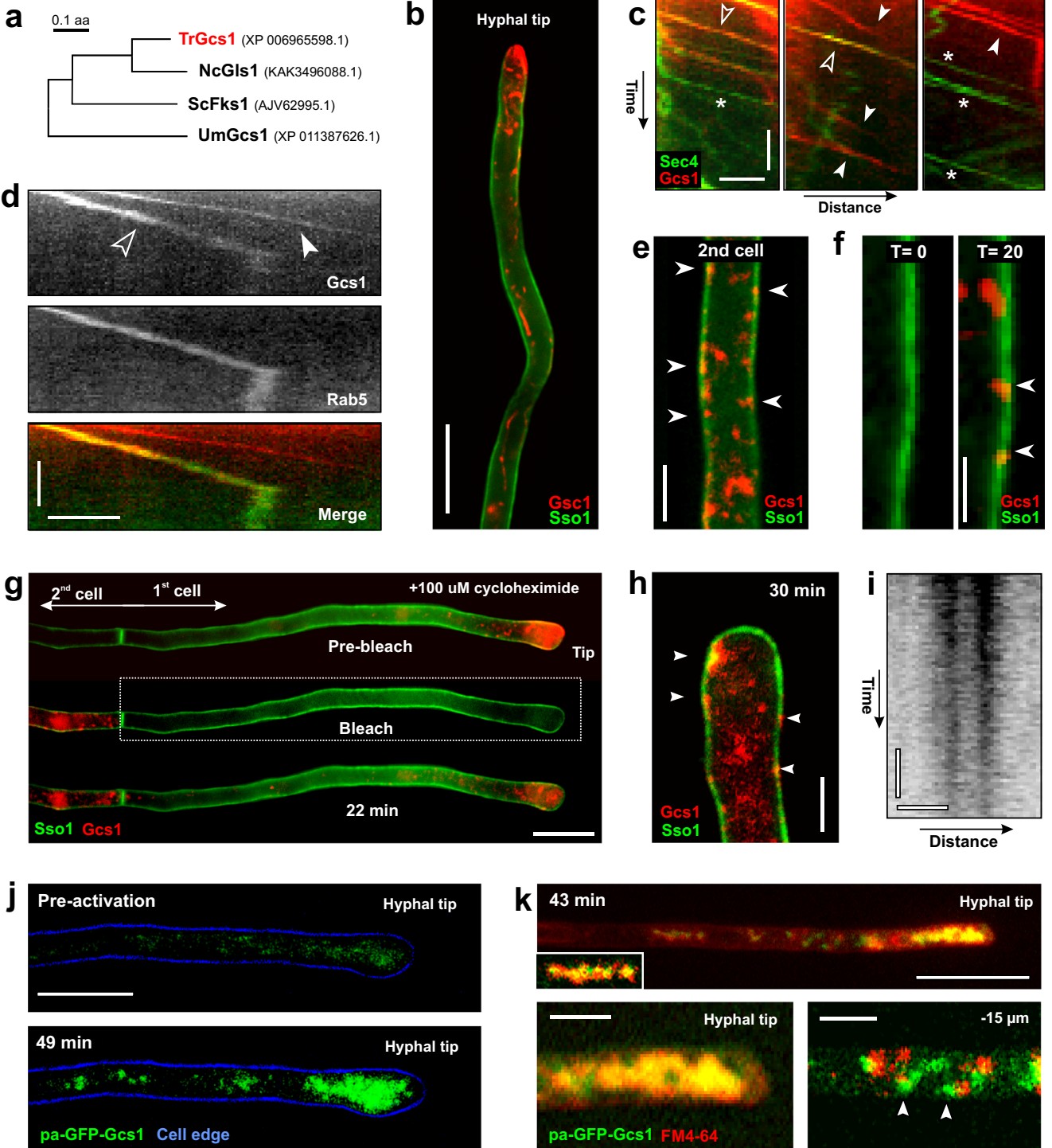

Our results pointed towards an important role of EE motility from sub-apical cells to the hyphal tip, which most likely delivers cell wall-synthases and other growth supplies to the expanding apex. We finally set out to test the importance of EEs motility for hyphal growth in *T. reesei*. Previous work has identified hook-like proteins as adaptors for molecular motors on fungal EEs[67,68]. We identified a hook protein in the *T. reesei* genome (TrHok1; Fig. 9a and Supplementary Table 4) and deleted the gene in a strain that expressed TrmsGFP-Rab5 (strain QM6a_ΔHok1_TrmsGRab5). In the absence of *Trhok1* almost all directed EE motility ceased (Fig. 9b and Supplementary Fig. 10), confirming that TrHok1 is a motor adaptor in *T. reesei*. Hyphae of QM6a_ΔHok1 were viable, albeit were growing significantly slower than wild-type

hyphae (Fig. 9c) and showed increased branching in subapical cells (Fig. 9d; leading hyphae indicated by open arrowhead; subapical branches indicated by closed arrowhead). Thus, we conclude that EE trafficking is not essential for cellular survival but supports the focus of the growth region at the leading hypha and increases its growth rate.

## Discussion

Hyphal growth enables the otherwise immobile fungus to overcome distances. As such, it is of central importance for all aspects of fungal biology. The current understanding is that hyphal growth is mediated by the apical 1st cell, which expands at the tip[7–10,69] to invade substrates, soil or living tissue. Subapical compartments, separated from the

**Fig. 8 | Delivery of 1,3-β-glucan synthase from subapical cells to the hyphal tip.**
**a** Maximum likelihood tree of fungal 1,3-β-glucan synthases. NCBI accession numbers given in parenthesis; tree was generated using MEGA5.2[73]. **b** Localisation of fluorescent 1,3-β-glucan synthase (TrmCherry₂-Gcs1) in a hypha; plasma membrane labelled with TrmsGFP-Sso1 (green). **c** Kymographs showing anterograde motility of SVs (TrmsGFP₂-Sec4; green) and 1,3-β-glucan synthase (TrmCherry₂-Gcs1; red); open arrowheads: 1,3-β-glucan synthase in SVs, closed arrowheads: 1,3-β-glucan synthase travelling independent of SVs; asterisk: SVs travelling independent of 1,3-β-glucan synthase. See also Supplementary Fig. 9a, Supplementary Movie 15. Horizontal scale bar = 2 μm; vertical scale bar = 2 s. **d** Kymographs showing anterograde motility of TrmsGFP-Rab5 (EEs; green in Merge image) and 1,3-β-glucan synthase (Gcs1; red in Merged image); open arrowheads: 1,3-β-glucan synthase in EEs (yellow); closed arrowheads: 1,3-β-glucan synthase travelling independent of EEs; Horizontal scale bar = 3 μm; vertical scale bar = 3 s. See also Supplementary Fig. 9b, Supplementary Movie 16 and 17. **e** Localisation of TrmCherry₂-Gcs1 (red) in the plasma membrane (green, labelled with TrmsGFP-Sso1) of a subapical 2nd cell. Arrowheads indicate red signals in the plasma membrane. Scale bar = 3 μm. **f** Re-appearance of TrmCherry₂-Gcs1 (red) in the plasma membrane (green, TrmsGFP-Sso1) of a 2nd cell after photobleaching. Time in minutes after laser-treatment is given above panels; inserted red signals are indicated by arrowheads; the green signal was taken prior to photo-

bleaching. Scale bar = 2 μm. **g** FRAP of TrmCherry₂-Gcs1 at the hyphal apex; plasma membrane labelled with TrmsGFP-Sso1 (green); box indicates photo-bleached region; time before (Pre-bleach), at laser treatment (Bleach) and after recovery time (22 min) is indicated. Sale bar = 10 μm. **h** Recovered TrmCherry₂-Gcs1 in the hyphal apex of a photo-bleached 1st cell. Arrowhead indicates plasma membrane location. Recovery time after bleaching is indicate. Scale bar = 3 μm. **i** Contrast-inverted kymograph of recovered TrmCherry₂-Gcs1 signals in the apical plasma membrane of a photo-bleached 1st cell; stationary signals indicate enzyme activity[66]. Horizontal scale bar = 1 μm; vertical scale bar = 3 s. **j** Accumulation fluorescent paGFP-Gcs1 (green) in the hyphal apex. Before photo-activation, no signal is detected (Pre-activation); 49 min after photo-activation in the 2nd cell, fluorescent paGFP-Gcs1 accumulates at the tip. The cell edge is shown in blue. Scale bar = 10 μm. **k** Co-localisation of the endocytic marker dye FM4-64 (red) and paGFP-Gcs1 (green) at a hyphal tip at 43 min after activation in the 2nd cell. Subpanels show co-localisation in the hyphal apex (left); few green signals (arrowheads) indicate that paGFP-Gcs1 is also delivered in vesicles that have not received FM4-64 (e.g., recycling vesicles, formed before FM4-64 treatment or SVs). Scale bars = 10 μm (overview) and 3 μm (lower panels). Results shown in (**b**–**k**) were confirmed in 2 independent experiments. Protein synthesis in (**g**, **h**) is inhibited by 100 μM cycloheximide[65]. Data in (**b**–**f**) were obtained from growing hyphae and are provided in the Source Data file.

---

apical cell and each other by septa, appear to have no role in hyphal tip growth[20–23]. We show in this study that the subapical 2nd and 3rd cell are crucial for hyphal growth of *T. reesei*, and that the subsequent cells increase the speed of hyphal extension. These data support previous work that showed that subapical cells participate in tip growth in *Penicillium chrysogenum*, *Aspergillus niger* and *Aspergillus wentii*[24]. However, this early study focussed on growth of entire colonies and was contradicted by sophisticated laser dissection experiments, showing that the apical 1st cell is self-sustained in hyphal growth of *A. niger*, *P. chrysogenum* and other fungi[22,23]. Consistent with a sole function of the tip cell in growth, live cell imaging of WBs in *A. niger* and *A. oryzae* showed that 40–50% of the first three septa are "plugged", while incidence of closure of more subapical septa is even higher[26,27]. This strongly reduces inter-cellular exchange of cytoplasm and organelles. In contrast, we found that > 85% of the first five septa in *T. reesei* are open and, therefore, could allow subapical support of tip growth[24].

It was recently shown that hyphae of two ascomycetes and one basidiomycete only need their tip cell for hyphal expansion[22,23]. In the light of this, *T. reesei* is exceptional in using subapical cells to support growth. From a functional point of view, these subapical compartments fall into two categories. Firstly, the subapical 2nd and 3rd cell are essential for tip expansion, which correlates with CS in these compartments. In particular, the 2nd cell is of importance, as it also provides more anterograde cross-septum EE trafficking, contains more Golgi cisternae and more ER tubules. Together with the tip cell, the 2nd cell, and to a lesser degree the 3rd cell, form a "core growth unit" (CGU; Fig. 9e), that, like the apical 1st cells in *A. niger* and *P. chrysogenum*[22,23], is self-sustained in hyphal growth. In all 3 fungi, this CGU extends at similar rates (77–112 μm h⁻¹) and has a comparable length (243–363 μm; Supplementary Table 3). However, considering the volume of the CGU, *T. reesei* uses 2-times and 4-times less cytoplasm to extend the CGU by 1 μm h⁻¹ than *P. chrysogenum* and *A. niger*, respectively (Supplementary Table 3, see V/G ratio). The second category of subapical cells in *T. reesei* covers the 4th- 6th cell and probably subsequent cells, not analysed in this study. These subapical compartments increase the rate by which hyphae extend. Collectively, they raise the growth rate from 77 μm h⁻¹ (the CGU alone) to ~190 μm h⁻¹ (the intact hypha). Thus, like an "engine booster", these subapical cells increase the performance of the hypha, thereby enabling *T. reesei* to travel distances effectively. Such a mechanism is not found in the other fungi, where the hyphal growth speed of intact hyphae is not exceeding that of the apical cell alone (Supplementary Table 3). Thus, we conclude that *T. reesei* is exceptionally efficient in mediating hyphal growth.

It is widely accepted that hyphal tip expansion is driven by cellular turgor pressure. This hydrostatic force is a consequence of water uptake into the hyphal cell, which is driven by osmotically active solutes inside the cytoplasm[25,39]. The resulting force is directed to the growing apex, where the cell wall is relatively flexible[8]. As a consequence, cytoplasm and embedded organelles drift towards the expanding hyphal tip[39,40]. We found that most EEs show such anterograde drift, which was used as an indication of CS. Surprisingly, tip-directed CS was largely restricted to the 1st, 2nd and 3rd cell and declined in the 4th and subsequent subapical cells. This suggests that the force for tip-directed CS and polar hyphal growth is generated in the 3 cells that form the CGU in *T. reesei*. It is not clear why the internal pressure is restricted to the first 3 cells, as we show that the septum between the 3rd and 4th cell is permeable.

Our quantitative analysis of EE motility revealed a net flow of endocytic membranes from subapical compartments towards the hyphal tip. This suggests that endocytic recycling occurs in subapical cells. Indeed, this study provides evidence that endocytosis occurs in all subapical cells. Firstly, we found that the endocytic marker dye FM4-64 accumulates in the vacuolar membrane of the 1st and all subapical cells. The vacuole is the terminal endocytic compartment[47,53], strongly suggesting that endocytosis occurs in all cells. Secondly, we visualised the formation of endocytic transport vesicles at the plasma membrane by monitoring the dynamic behaviour of F-actin patches[54–56]. We found dynamic actin patches in the 1st and all subapical cells. Quantitative analysis of actin patch distribution revealed that each subapical cell shows 25–30% of the endocytic activity of the entire 1st cell (Table 1). When the difference in cell length is not considered and instead 1 μm² areas of the plasma membrane are compared (see above, Fig. 6f), the endocytic activity in subapical cells reaches even 38–45% of that in the apex of the 1st cell. Thus, the subapical cells show high endocytic activity. These results challenge the current understanding that the apex of the 1st cell is the only part of the hypha that undergoes significant endocytosis[16,18,19].

Subapical endocytosis and cell-to-cell transport of EEs towards the growing tip cell requires a mechanism by which subapical cells replenish recycled membranes and proteins. We speculated that subapical exocytosis maintains the membrane homoeostasis in subapical cells. Exocytosis at the plasma membrane involves SNARE proteins at the site of vesicle fusion[61]. Previous observation of this trans-SNARE complexes supports the notion that subapical exocytosis occurs in *T. reesei*[35]. We visualised endogenous levels of Snc1, a homologue of the vesicle-associated v-SNARE synaptobrevin[35], in growing hyphae of *T. reesei*. We report that v-SNARE-positive vesicles are moving within the

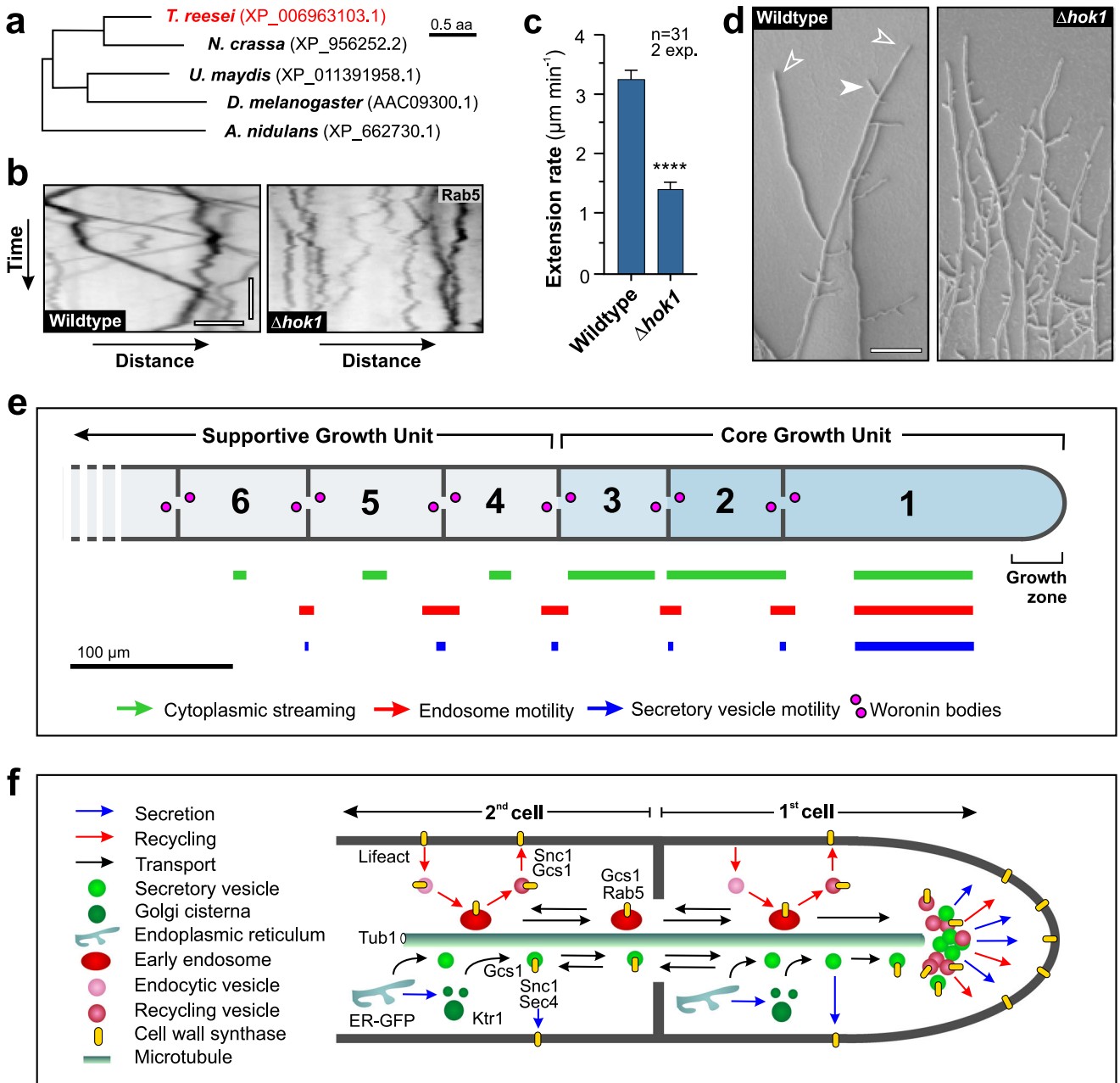

**Fig. 9 | The role of EE motility in hyphal growth and graphical summary.**
**a** Maximum likelihood tree of putative Hook adaptor proteins, reported to bind motor proteins to EEs[67,68]. NCBI accession numbers given in parenthesis; tree was generated using MEGA5.2[73]. **b** Contrast-inverted kymographs showing EE motility in wildtype strain QM6a and in a *hok1* deletion mutant. Horizontal scale bar = 2 μm; vertical scale bar = 3 s. See also Supplementary Fig. 10. **c** Hyphal growth rate in strain QM6a (Wildtype) and a *Trhok1* deletion mutant (Δ*hok1*). Sample size *n* = 31 hyphae from 2 independent experiments. **d** Morphology of wildtype strain QM6a and a *Trhok1* null mutant. Deletion of the endosomal motor adaptor gene *hok1* results in increased branching, which is probably a consequence of reduced hyphal growth rate. Open arrowheads indicate leading hypha, closed arrowhead indicates subapical branching. Cells were grown on agar plates. Scale bar = 100 μm. **e** Graph summarising the contribution of the apical and following 5 subapical cells in hyphal growth of *T. reesei* hyphae. Note that motility of SVs (blue bars) and EEs (red bars) through septa is given; drift of EEs due to CS (green bars) was measured within cells; the length of each bar represents the relative amount found in tip cells (20–30 μm behind the apex). The length of each cell is drawn to scale. **f** Model of the contribution of subapical secretion and endocytic recycling in tip extension. Marker genes, used to label compartments, vesicles, microtubules and cargo, are included. Results shown in (**b**, **c**) were confirmed in 2 independent experiments. Bars in (**c**) represent mean ± SEM; sample sizes are indicated. Statistical comparison with in (**c**) used Student's *t* testing testing; ****= two-tailed *P*-values < 0.0001. All data are provided in the Source Data file.

2[nd] cell before they fuse with the plasma membrane of this subapical compartment. Moreover, we performed FRAP-experiments in a 2[nd] cell, which revealed that 1,3-β-glucan synthase is secreted in this subapical compartment, where it may participate in the maturation of fungal cell walls in older parts of the hypha[70]. These findings are consistent with the existence of exocytosis and secretion in subapical exocytosis.

We found that all subapical compartments contain a prominent ER network and Golgi compartments. It was reported that Golgi compartments are concentrated in the hyphal tip cell, where they form SVs for apical exocytosis[10]. The presence of the Golgi in subapical cells supports the notion that exocytosis of newly synthesised membranes and proteins occurs in subapical cells. However, 1,3-β-glucan synthase was also shown to undergo endocytic recycling in *A. nidulans*[64].

Endocytic recycling involves EEs[47], and we show in this study that 1,3-β-glucan synthase is delivered from subapical cells to the hyphal tip in rapidly moving EEs. Thus, secretion and endocytic recycling of cell wall synthases in subapical cells, and the subsequent long-distance delivery of these enzymes to the apex of the 1st cell support the extension of hyphae. Whether other industrial enzymes, such as cellulases, are also secreted in subapical cells remains to be investigated.

In conclusion, our results demonstrate that subapical and tip cells cooperate in hyphal growth. The apical tip cell and the first 2 subapical cells are essential for hyphal growth; together they can be considered the "core growth unit" of the hypha (Fig. 9e). CS was restricted to these compartments, suggesting that bulk flow of cytoplasm from subapical cells is essential for hyphal tip growth, a notion that was previously suggested[24]. We also found evidence for a role of MTs in membrane trafficking across septa. A net flow of EEs from subapical cells to the hyphal tip, which delivers 1,3-β-glucan synthase, and presumably other growth supplies, to the expanding hyphal apex (Figs. 9e, f). This transport coincides with prominent endocytic recycling and secretion in subapical cells, which could supply the membranes and enzymes to the growing tip. That secretion and endocytosis occur in subapical cells challenges the current understanding of hyphal growth in filamentous fungi[16–19] but must be considered in order to understand growth but also industrial enzyme production in *Trichoderma reesei*.

Hyphae still grew at a reduced growth rate when EE motility was abolished (~ 50% reduction). We found a similar growth speed reduction when the 4th- 6th are damaged. Thus, it is possible that these subapical cells support growth by providing growth supplies, such as cell wall synthases, to the hyphal tip. We therefore consider the 4th- 6th cell, and probably additional distal cells, a "supportive growth unit" (SGU; Fig. 9e) that assists hyphal growth by providing newly synthesised and recycled growth supplies. While subapical cells and the tip cell cooperate in hyphal extension in *T. reesei*, other fungi, such as *A. niger* or *P. chrysogenum*, do not rely on subapical cells for hyphal growth[22,23]. Moreover, invading hyphae of the basidiomycete plant pathogen *Ustilago maydis* have no subapical cells, but consist of just a tip cell[71]. Thus, hyphal growth can be achieved by different mechanisms. Future studies are needed to gain a wider understanding of the role of subapical cells in protein secretion, endocytosis, and growth of filamentous fungi.

# Methods

## Biological material
**Fungal strains**. *T. reesei* wild type strain QM6a was generously provided by Prof. David Archer, Nottingham, UK, and can be obtained from American Type Culture Collection (https://www.atcc.org/; ATCC13631). All *T. reesei* strains were stored at − 80 °C as glycerol stocks (25% glycerol), and cultures were grown on PDA plates (24 g l⁻¹ potato dextrose broth (PDB); 20 g l⁻¹ agar; Sigma-Aldrich, Poole, UK) at 25 °C for 4–5 days. For all strains used in this study see Supplementary Table 1.

**Bacteria**. Plasmids were propagated in *Escherichia coli* strain DH5α (Thermo Fisher Scientific, UK). *A. tumefaciens*-mediated transformation of *T. reesei* used *A. tumefaciens* strain EHA105[72] (GoldBio, St Louis, USA). Bacteria were grown in double-strength yeast extract/tryptone DYT medium (tryptone, 16 g l⁻¹; yeast extract, 10 g l⁻¹; NaCl, 5 g l⁻¹; with 20 g l⁻¹ agar in plates) at 37 and 28 °C, respectively.

## Identification of *T. reesei* homologues and bioinformatics
To identify homologues of the chosen reporter proteins, we screened the published sequence of *T. reesei* (https://mycocosm.jgi.doe.gov/Trire2/Trire2.home.html), using the provided BLASTP function and the *Z. tritici* protein sequences of homologues of Sso1 (NCBI accession number: XP_003857391.1), Hex1 (NCBI accession number: XP_003854425.1), Rab5 (NCBI accession number: XP_003857299.1),

Tub1 (NCBI accession number: XP_003848766.1), Sec4 (NCBI accession number: XP_003854966.1), Gcs1 (NCBI accession number: XP_003848451.1), the *Saccharomyces cerevisiae* protein sequences of α −1,2-mannosyltransferase Krt1p (NCBI accession number: AJU11104.1), the exocyst subunit Exo70 (NCBI accession number NP_012450.1) and the *U. maydis* protein sequence of Hok1 (NCBI accession number: XP_761698.1). The homologue of *T. reesei* Snc1 protein sequence (NCBI accession number: AAT78419.1) was previously published[35]. Sequences were obtained from the NCBI server (https://www.ncbi.nlm.nih.gov/) and comparison was done using EMBOSS Needle (http://www.ebi.ac.uk/Tools/psa/emboss_needle/), and domain structures were analysed in PfamScan (https://www.ebi.ac.uk/Tools/pfa/pfamscan/).

## Molecular cloning
Plasmids (Supplementary Table 2) were generated by in vivo recombination in *S. cerevisiae* DS94 (MATα, *ura3-52*, *trp1-1*, *leu2-3*, *his3-111*, and *lys2-801*). *S. cerevisiae* DS94 cells were grown for 12 h in 3 ml YPD at 28 °C and 200 rpm and used to inoculate 50 ml YPD. After an additional 5 h at 28 °C, 200 rpm, cells were harvested by centrifugation at 672 × *g* for 5 min, washed with 5 ml sterile distilled water, and after re-centrifugation, re-suspended in 300 µl sterile water. For in vivo recombination, ~ 100 ng in 4 µl volume of each purified DNA fragment using silica glass suspension were mixed with 50 µl salmon sperm DNA (2 mg ml⁻¹ stock; Sigma-Aldrich), 50 µl (~ 15 million) *S. cerevisiae* DS94 cells, 32 µl 1 M lithium acetate, and 240 µl 50% (v v⁻¹) PEG 4000 (Sigma-Aldrich). Samples were gently mixed by pipetting up and down and incubated at 28 °C for 30 min. A heat shock was induced at 45 °C for 15 min, and samples were centrifuged at 400 × *g* for 2 min at room temperature. Sedimented cells were re-suspended in 150 µl sterile distilled water and were plated onto yeast synthetic drop-out medium, which lacks uracil (yeast nitrogen base without amino acids and ammonium sulphate, 1.7 g l⁻¹; ammonium sulphate, 5 g l⁻¹; casein hydrolysate, 5 g l⁻¹; tryptophan, 20 mg l⁻¹; agar, 20 g l⁻¹), followed by 2 d incubation at 28 °C. Colony PCR was performed on yeast cells by using DreamTaq DNA polymerase (Thermo Scientific, Leicestershire, UK) in a 20 µl total volume. Plasmid DNA was isolated from the positive yeast colonies and transformed into and amplified in *E. coli* strain DH5α. Finally, the plasmid DNA was isolated from the *E. coli* colonies and further confirmed by restriction analysis. All restriction enzymes were obtained from New England Biolabs (Ipswich, UK). For technical details on plasmid generation see Supplementary Methods.

## *T. reesei* transformation
All plasmids generated in this study were transformed into *T. reesei*, using *A. tumefaciens* EHA105-mediated transformation[72]. In brief, plasmids were introduced into *A. tumefaciens* by heat shock and transformants were grown overnight at 28 °C at 200 rpm in 10 ml DYT medium supplemented with 20 µg ml⁻¹ rifampicin (Melford, Ipswich, UK) and 50 µg ml⁻¹ kanamycin. The overnight cultures were diluted to an optical density of 0.15 at 660 nm in *Agrobacterium* induction medium AIM, supplemented with 200 µM acetosyringone (Sigma−Aldrich, Gillingham, UK) and grown at 28 °C and 200 rpm until an optical density of 0.3–0.35 (3–4 h). The *A. tumefaciens* cultures were then mixed with an equal volume of *T. reesei* conidia, which had been harvested from 5 day old PDA plates, grown at 25 °C, and diluted to a concentration of 1 × 10⁶ ml⁻¹ in *Agrobacterium* induction medium. 200 µl of the *A. tumefaciens* − *T. reesei* mixtures were plated onto nitrocellulose filters (AA packaging limited, Preston, UK), placed on AIM agar plates supplemented with 200 µM acetosyringone and grown at 25 °C for 2 days. Nitrocellulose filters were then transferred onto PDA plates (Oxoid, Basingstoke, UK) containing 100 µg ml⁻¹ cefotaxime (Melford, Ipswich, UK), 100 µg ml⁻¹ timentin (Melford, Ipswich, UK) and either 100 µg ml⁻¹ carboxin (Sigma-Aldrich, Gillingham, UK) or 100 µg ml⁻¹ hygromycin B (Roche, West Sussex, UK) and incubated at 25 °C for 6–8 days. The individual colonies were transferred to PDA

plates containing 100 µg ml⁻¹ cefotaxime, 100 µg ml⁻¹ timentin, and either 100 µg ml⁻¹ carboxin or 100 µg ml⁻¹ hygromycin B and grown at 25 °C for 4–5 days. Plasmids pTrCTrmsGFPSso1, pTrCTrmsGFPRab5, pTrCLifeactTreGFP, and pTrCCalᵟTreGFPHDEL were integrated into the *sdi1* locus of *T. reesei* QM6a, resulting in strains QM6a_TrmsGSso1, QM6a_TrmsGRab5, QM6a_LifeactTreG, and QM6a_CalᵟTreGFPHDEL, respectively. To visualise Golgi vesicles, plasmid pTrHKtr1TreGFP was integrated into the *ktr1* locus in strain QM6a, resulting in QM6a _Ktr1TreG. To co-visualise EEs and plasma membrane, cytoplasm and plasma membrane, the plasmid pTrHTrmCherrySso1 was random ectopically integrated into strains QM6a_TrmsGRab5 and QM6a_TrmsG, resulting in strains QM6a_TrmsGRab5_TrmChSso1 and QM6a_TrmsG_TrmChSso1, respectively.

Plasmid pTrHTrmCherrySso1 was random ectopically integrated into strain QM6a, resulting in QM6a_TrmChSso1. To co-visualise WBs and plasma membrane, microtubules and plasma membrane, mitochondria and plasma membrane, plasmids pTrCHex1TrmsGFP and pTrCTrmsGFPTub1 were integrated into the *sdi1* locus of *T. reesei* QM6a_TrmChSso1, resulting in strains QM6a_TrmChSso1_Hex1TrmsG and QM6a_TrmChSso1 _TrmsGTub1, respectively. To co-visualise SVs and plasma membrane, first plasmid pTrHTrmsGFP₂Sec4 was integrated into the *sec4* locus strain QM6a, resulting in QM6a_TrmsG₂Sec4. Then plasmid pTrCTrmCherrySso1 was integrated into the *sdi1* locus of strain QM6a_TrmsG₂Sec4, resulting in QM6a_TrmsG₂Sec4_TrmChSso1. To co-visualise synaptobrevin and plasma membrane, plasmid pTrCTrmsGFPSnc1 was integrated into the *snc1* locus of strain QM6a_ TrmChSso1, resulting in QM6a_ TrmChSso1_TrmsGSnc1. To co-visualise synaptobrevin and SVs, plasmid pTrCTrmCherrySnc1 was integrated into the *snc1* locus of strain QM6a_TrmsG₂Sec4_resulting in QM6a_TrmsG₂Sec4_TrmChSnc1. To visualise exocyst, plasmid pTrHTreGFPExo70 was integrated into the *exo70* locus in strain QM6a, resulting in QM6a _TreGExo70. To co-visualise 1,3-β-glucan synthase and plasma membrane, first, the plasmid pTrCTrmCherry₂Gcs1 was integrated into the *gcs1* locus strain QM6a resulting in QM6a_TrmCh₂Gcs1. Then, plasmid pTrHTrmsGFPSso1 was random ectopically integrated into the strain QM6a_TrmCh₂Gcs1, resulting in QM6a_TrmCh₂Gcs1_TrmsGSso1. To co-visualise SVs and 1,3-β-glucan synthase, plasmid pTrCTrmCherry₂Gcs1 was integrated into the *gcs1* locus of strain QM6a_TrmsG₂Sec4, resulting in QM6a_TrmsG₂Sec4_TrmCh₂Gcs1. To co-visualise 1,3-β-glucan synthase and EEs, plasmid pTrHTrmsGFPRab5 was random ectopically integrated into the strain QM6a_TrmCh₂Gcs1, resulting in QM6a_TrmCh₂Gcs1_TrmsGRab5. To co-visualise photo-activatable 1,3-β-glucan synthase and plasma membrane, plasmid pTrCpaGFPGcs1 was integrated into the *gcs1* locus of strain QM6a_TrmChSso1, resulting in QM6a _TrmChSso1_paGGcs1. Plasmid pTrHΔHok1 was integrated into the *hok1* locus of QM6a, resulting in strain QM6a_ΔHok1. To visualise EEs in the *hok1* null mutant, plasmid pTrCTrmsGFPRab5 was integrated into the *sdi1* locus of strain QM6a_ΔHok1, resulting in QM6a_ΔHok1_TrmsGRab5.

## Molecular analysis of transformants

Correct integration of plasmids into the genome of *T. reesei*, was confirmed by PCR using DreamTaq DNA polymerase (Thermo Scientific, Leicestershire, UK) in 20 µl total volume. To this end, *T. reesei* transformants were grown on PDA, supplemented with either 100 µg ml⁻¹ carboxin or 100 µg ml⁻¹ hygromycin B for 4–5 days at 25 °C. A loopful of fungal mycelium (collected with pipette tip) was added to 1.5 ml Eppendorf tubes containing 200 µl TE buffer (10 mM Tris HCl; 1 mM EDTA, pH-8.0) and 0.3 g acid-washed glass beads (425–600 µm diameter, Sigma–Aldrich, Gillingham, UK). The tubes were vortexed for 30 min using an IKA Vibrax shaker (IKA, Staufen, Germany). 1 µl of this mixture was used for PCR reactions. Integration of plasmids into the *sdi1* locus was confirmed with primers SK-Tri-455 and SK-Tri-453. The expected band sizes are: 1.2 kb for wild type, 5.3 kb for

plasmidpTrCTrmsGFPSso1, 4.9 kb for plasmid pTrCHex1TrmsGFP, 4.9 kb for plasmid pTrCTrmsGFPRab5, 6.0 kb for plasmid pTrCTrmsGFPTub1, 4.1 kb for plasmid pTrCLifeactTreGFP, 4.1 kb for pTrCCalᵟTreGFPHDEL and 5.3 kb for plasmid pTrCTrmCherrySso1. Integration of the plasmid into the *sec4* locus was confirmed with primers SK-Tri-136 and SK-Tri-141. The expected band sizes are: 2.0 kb for wild type and 3.4 kb for plasmid pTrHTrmsGFP₂Sec4. Integration of the plasmid into the *ktr1* locus was confirmed with primers SK-Tri-454 and SK-Tri-445. The expected band sizes are: 1.4 kb for wild type and 4.0 kb for plasmid pTrHKtr1TreGFP. Integration of plasmids into the *snc1* locus was confirmed with primers SK-Tri-586 and SK-Tri-587. The expected band sizes are: 3.1 kb for wild type and 7.0 kb for plasmid pTrCTrmCherrySnc1 and pTrCTrmsGFPSnc1. Integration of the plasmid into the *exo70* locus was confirmed with primers SK-Tri-326 and SK-Tri-327. The expected band sizes are: 3.5 kb for wild type and 5.7 kb for plasmid pTrHTreGFPExo70. Integration of plasmids into the *gcs1* locus was confirmed with primers SK-Tri-91 and SK-Tri-102. The expected band sizes are: 2.3 kb for wild type, 3.7 kb for plasmid pTrCTrmCherry₂Gcs1 and 3.0 kb for plasmid pTrCpaGFPGcs1. Integration of the plasmid into the *hok1* locus was confirmed with primers SK-Tri-221 and SK-Tri-222. The expected band sizes are: 5.0 kb for wild type and 3.7 kb for plasmid pTrHΔHok1. PCR analysis of *T. reesei* transformants revealed that targeted integration of plasmids into the *sdi1*, *sec4*, *ktr1*, *snc1*, *gcs1*, and *hok1* loci yielded both the wild-type and mutant bands. To overcome this obstacle, conidia were harvested from 5-day old PDA plates containing either 100 µg ml⁻¹ carboxin or 100 µg ml⁻¹ hygromycin B, and the number of conidia was determined using a Cellometer Auto 1000 cell counter (Nexcelom Biosciences, Lawrence, USA). A dilution series was prepared (0.01 × 10⁶ ml⁻¹, 0.001 × 10⁶ ml⁻¹, 0.001 × 10⁶ ml⁻¹), and 100 µl of this cell suspension were placed on the PDA plates with either 100 µg ml⁻¹ carboxin or 100 µg ml⁻¹ hygromycin B, followed by incubation for 2–3 days at 25 °C. The individual colonies were further transferred onto PDA plates containing either 100 µg ml⁻¹ carboxin or 100 µg ml⁻¹ hygromycin B and grown at 25 °C for 4–5 days. Finally, PCR was performed as described above to confirm the homokaryotic transformants.

**Transmission electron microscopy** was performed by fixing PDB-grown cells with 2% (w v⁻¹) glutaraldehyde and 2% (w v⁻¹) paraformaldehyde in 0.1 M PIPES buffer (pH 7.2) for 2 h at room temperature followed by 3 × 5 min washes in the same buffer before post-fixation in 2% (w v⁻¹) potassium permanganate in deionised water for 1 h on a rotator. After 3 × 5 min washes in ddH₂O, the cells were dehydrated in a graded ethanol series of 30, 50, 70, 80, 90, 95% ethanol for 10 min per step, followed by 2 × 20 min in 100% ethanol and subsequently embedded in SPURR resin. Ultrathin sections (60 nm) were collected on pioloform-coated 100 mesh copper EM grids (Agar Scientific, Stansted, UK) and contrasted using Reynold's lead citrate for 10 min before imaging using a JEOL JEM 1400 transmission electron microscope operated at 120 kV. Images were taken with a digital camera (ES1000W, Gatan, Abingdon, UK).

**Scanning electron microscopy** *T. reesei* strains QM6a and QM6a_ΔHok1 were grown on PDA for 2 days at 25 °C. The 10 mm diameter agarose discs containing hyphal cells were attached to a cryostage and frozen rapidly in liquid nitrogen slush. Water sublimation at −95 °C for 3 min was performed using a Jeol JSM-6390LV scanning electron microscope (JEOL, Ltd, Welwyn Garden City, UK). This was followed by gold sputter coating using an Alto 2100 chamber (Gatan Ltd., Oxfordshire, UK) and observation in a Jeol JSM-6390LV scanning electron microscope (JEOL, Ltd)

**Fluorescence microscopy** was done using a motorised inverted microscope (IX83; Olympus, Hamburg, Germany), equipped with a PlanApo 100 ×/1.45 Oil TIRFM or UPlanSApo 60X/1.35 Oil objective (Olympus). Fluorescent proteins or fluorescent dyes were excited using a VS-LMS4 Laser Merge System with 75 mW solid-state lasers (488 nm, 561 nm; Visitron Systems, Puchheim, Germany) or an HBO

mercury lamp. For photo-bleaching, photo-activating, and laser-induced rupture experiments, a 405 nm/60 mW diode laser was which was coupled into the light path by an OSI-IX 71 adaptor (Visitron System) and controlled by a UGA-40 unit and a VisiFRAP 2D FRAP control software (Visitron System) was used. A Piezo objective (Piezosystem Jena GmbH, Jena, Germany) was used to generate Z-axis image stacks. Dual observation of red and green fluorescence was performed using a dual imager (Dual-View 2 Multichannel Imaging System; Photometrics, Tucson, USA) equipped with a dual-line beam splitter (z491/561; Chroma Technology Corp., Bellows Falls, USA), with an emission beam splitter (565 DCXR; Chroma Technology Corp.), an ET-Band pass 525/50 (Chroma Technology Corp.), and a single band pass filter (BrightLine HC 617/73; Semrock, New York, USA). Images were captured using a PRIME-BSI-Express camera (Photometrics). All parts of the system were under the control of the software package VisiView (Visitron System).

**Total internal reflection fluorescence microscopy (TIRFM)** was done using a motorised inverted microscope (IX83; Olympus), equipped with a VisiTIRF Dual port condenser VS T1 (Visitron Systems) and a PlanApo 100 ×/1.45 Oil TIRF objective (Olympus). Fluorescent proteins or fluorescent dyes were excited using a VS-LMS4 Laser Merge System with 75 mW solid-state lasers (488 nm, 561 nm; Visitron Systems). Images were captured using a CoolSNAP HQ2 CCD camera (Photometrics). All parts of the system were under the control of the software package VisiView (Visitron System). MetaMorph 7.8.x (Molecular Devices, Wokingham, UK) was used for all image processing.

**Visualisation of cytoskeleton, septa, vesicles and organelles** was done in *T. reesei* strains grown at 25 °C for 12 h on PDA plates or at 25 °C with 100 rpm for 12 h in PDB liquid media. From the PDA plates a 1 × 1 cm area of the colony edge was cut and placed on a microscope slide. A drop of PDB was added onto the colony before covering with a coverslip. From the PDB liquid cultures, 1 μl was placed onto a 3% agar cushion, followed by microscopic observation. Acquisition of Z-axis image stacks were acquired with a Z resolution of 0.2 μm and an exposure time of 150 ms. The 488 nm laser was used at 10–50% intensity. To acquire quantitative data of EE drift or movement of organelles over the septum a single image of the labelled plasma membrane was taken as a reverence followed by acquisition of streams with a frame rate of 6.66 fps to 12 fpm and a laser intensity between 10–100%. For example images and movies of organelle movement over the septum a single image of the Sso1 labelled septa was taken followed by photo bleached an area of interest using the 405 nm laser 30% intensity. A movie of the moving organelles was taken with an exposure time of 150 ms and laser intensities of 20- 100%. Co-localisation of Gcs1 with Rab5 or Sec4 was done by photo bleaching on the area of interest using the 405 nm laser at 30% intensity, followed by acquisition of a movie of the red and green fluorescent protein fluorescence using the Dual-View 2 Multichannel Imaging System (Photometrics). Overlays were generated using MetaMorph. The abundance of Golgi vesicles and endoplasmic reticulum was analysed in sum-projections of Z-axis image stacks of strain QM6a _Ktr1-TreG or QM6a_TrG-HDEL. The average intensity was measured in the apical 10 μm of the 1st cell and in the middle region of cell 1–6. All measurements were corrected for the image background. The data sets were tested for outliers using the Grubbs test with an alpha of 0.2 using Prism10 (GraphPad Software, San Diego, USA). Identified outliers were removed from the graphs but are highlighted in the source data file. For quantitative analysis of actin patch numbers, the single top-view focal plane of Z-axis image stacks were chosen, and the number of patches in a region of interest was counted, and this number was correlated to an area of 1 μm². The analysis was done in the apical 5 μm, the middle region, and - 10 μm before the septum of the 1st cell, as well as in the middle region of each sub-apical cell. TIRF microscopy was used to visualise synaptobrevin dynamics at the plasma membrane in strain QM6a_TrmChSso1_TrmsGSnc1. The signal of the TrmChSso1 was used to find the correct focal plane and TIRF angle. This was followed by acquiring movies in the GFP channel using the 488 nm laser at 70% intensity with an exposure time of 150 ms.

## Inhibitors and dyes used in this study

The cytoskeleton inhibitors benomyl and latrunculin A (Molecular Probes/Invitrogen, Paisley, UK) were used at a final concentration of 10–12 μM (Stocks 10 mM DMSO), and cells were treated for 30 min before imaging. To visualise MT re-growth after depolymerisation, cells were incubated for 30 min in medium, supplemented with 12 μM benomyl. After 2 cycles of centrifugation at 2.4 g and subsequent rinsing with fresh PDB, cells were imaged as described. The protein synthesis inhibitor cycloheximide was used at a final concentration of 100 μM (Sigma-Aldrich, stock solution 100 mM DMSO). Cells were treated for 1 h before performing secretion experiments. Treated cells were placed onto a 3% agar cushion containing the corresponding inhibitor at the same concentration, followed by microscopic observation. Septa in strains QM6a_TrmsGSso1 and QM6a_LifeactTreG were counterstained with 5 μl ml⁻¹ Calcofluor White (Sigma, stock solution 1 mg ml⁻¹ in phosphate buffer, pH 7.0) for 10 min before imaging using the DAPI filter set and a HBO mercury lamp an exposure time of 1000 ms and the 488 nm laser at 20% intensity and an exposure time of 150 ms. To investigate endocytosis, cells of QM6a were stained with the endocytic marker dye FM4-64 (Stock 1 mM, Molecular Probes/Invitrogen, Paisley, UK). Cells were incubated in PDB containing 1 μM FM4-64 for 10 min, then washed by centrifugation for 5 min at 2.4 g and re-suspended in fresh PDB. Cells were incubated for 30 min on a SB2 Rotator (Bibby Scientific Limited, Stone, UK). Vacuoles were counter stained with 10 μM 5-(and-6)-carboxy-2',7-dicholorofluorescein diacetate (carboxy-DCFDA, Stock 10 mM, Thermo Fisher Scientific, Loughborough, UK) for 5 min before cells were imaged using the 488 nm and the 561 nm lasers at 10 and 50% intensity, respectively, and an exposure time of 150 ms.

## Assessment of growth and laser-induced cell rupture

Cells of strain QM6a_TrmsGSso1 were grown for 12 h on PDA. A 1 × 1 cm² area of the colony edge was cut and placed on a microscope slide. A drop of PDB was added onto the colony before covering with a coverslip. A hyphal tip was identified, and two DIC images were taken 5 min apart. If the hyphae were growing, the subapical cells were identified in the green channel using the 488 nm laser at 10% intensity. Laser-induced wounding was performed using a 500 ms light pulse of the 405 nm / 60 mW diode laser at 100% intensity. A DIC image of the hyphal tip was taken directly after and 45 min after wounding. Overlays of those images were used to analyse the ability to grow after laser-induced wounding.

## Analyses of retrograde EE displacement as an indication of loss of cytoplasm

To visualise retrograde EE displacement after laser-induced wounding of the second cell, strain QM6a_TrmsGRab5_TrmChSso1 was grown for 12 h on PDA, and microscope slides were prepared as described above. Growing hyphal tips were identified by taking two DIC images 5 min apart. The second cell was identified in the red channel using the 561 nm laser at 40% intensity. A movie of the GFP-labelled EE was acquired with a frame rate of 20–50 ms using the 488 nm laser at 20% intensity. During the movie acquisition, the second cell was wounded using a 500 ms light pulse of the 405 nm / 60 mW diode laser at 100% intensity. A DIC image of the hyphal tip was taken directly after and 45 min after wounding. Overlays of those images were used to analyse the ability to grow after laser-induced wounding. Retrograde EE displacement was analysed in kymographs generated in MetaMorph.

## Drift of endosomes as indicator of cytoplasmic streaming

To visualise retrograde EE drift close to the septa within the first 6 cells, strain QM6a_TrmsGRab5_TrmChSso1 was grown for 12 h on PDA. Sub apical cells in growing hyphae were identified in the red channel using the 561 nm laser at 40% intensity. Movies in the green channel with a frame rate of 6.66 fps and the 488 nm laser at 20% intensity were taken, and the EE displacement was analysed in kymographs generated in MetaMorph.

## Diffusion of cytoplasmic GFP over septa

Strain QM6a_TrmsG _TrmChSso1 was grown for 12 h on PDA. Microscope slides were prepared as described above, and the 1st and 3rd septa of growing hypha were identified in the red channel using the 561 nm laser at 40% intensity. A green image, with the 488 nm laser at 20% intensity, was taken before photo-bleaching, then either the cell before or behind the septa was bleached using a 405 nm laser at 30% intensity. Recovery was recorded immediately at 5 s intervals. To measure fluorescent recovery, the average intensity in the bleached area was corrected for the adjacent background and compared to a background-corrected measurement of the unbleached regions of the neighbouring cell. The average intensity over time in these regions was measured, and the percentage intensity of the bleached region relative to the unbleached region was calculated.

## Delivery and secretion of 1,3-β-glucan synthase from subapical cells

Gcs1 strain QM6a_TrmCh$_2$Gcs1_TrmsGSso1 was treated for 1 h with 100 μM of the protein synthesis inhibitor cycloheximide to visualise delivery and secretion. Corresponding images in the red and green channels were taken prior to photo-bleaching the entire 1st cell using the 405 nm laser at 30% intensity. Successful bleaching was confirmed by microscopy. Images of the entire 1st cell were taken 20–30 min post-bleaching using the 561 nm laser at 40% intensity. Delivery of photo-activatable glucan synthase was done in strain QM6a _TrmChSso1_paGGcs1. An image of the 1st cell before photo-activation was taken, followed by photo-activation of the 2nd cell, using the 405 nm laser at 5% intensity. After an additional 40–50 min, images of the apical region of the 1st cell hyphal tip were taken using the 488 nm laser at 30% intensity. Vesicles that contain endocytosed paGGcs1, which was photo-activation in the 2nd cell, using the 405 nm laser at 5% intensity, was co-stained by adding 1 μl of the endocytic marker dye FM4-64 to the 3% agar cushion. Cells were imaged after 43 min using the 488 nm laser at 30% and the 561 nm laser at 20% intensity.

To visualise secretion of TrmCh$_2$Gcs1 in the 2nd cells, images were taken at 10 μm behind the first septum of the 2nd cell prior and after photo-bleaching (405 nm laser at 30% intensity). Successful bleaching was confirmed afterwards, followed by taking images of TrmCh$_2$Gcs1 after an additional 20 min.

## Statistical analysis and data presentation

All measurements were done in raw 16-bit images using MetaMorph 7.8 (Molecular Devices). Data calculation was performed in Excel (Microsoft, Redmond, USA) or Prism10 (GraphPad Software, San Diego, USA), and all statistical testing was done using Prism10. Data sets with a sample size ≤ 6 were assumed to be normal distributed and are shown as mean ± standard error of the mean. For those datasets unpaired two-tailed Student's $t$ testing with Welch's correction or one-way ANOVA were used for statistical comparison. All data sets with a sample size $n > 6$ were tested for normal distribution using Shapiro-Wilk testing. In case at least one data set did not pass the normality test ($P < 0.05$), the data were presented as Whiskers' plots, with 25th/75th percentiles indicated as blue lines, median as a red line, and minimum and maximum values as whiskers ends. These data sets were tested using a nonparametric two-tailed Mann-Whitney test. All test results

are included in the figure legends. Full statistical information, including median, 25th/75th percentile, minimum/maximum values, are provided in the Source Data file. All graphs were generated in Prism10 and modified in CorelDraw X6 (Corel Corporation, Ottawa, Canada). Acquired images were adjusted in brightness, and contrast, and gamma, using MetaMorph.

## Reporting summary

Further information on research design is available in the Nature Portfolio Reporting Summary linked to this article.

## Data availability

The authors confirm that all relevant data are included in the paper or in the Supplementary Information file. The source data underlying Figs. 1a–k, 2a–e, 3a–g, 4a–f, 5a–f, 6a–f, 7a–h, 8b–k, 9b–d and Supplementary Figs. 2, 3a, b, 4a–c, 5a–c, 6, 7, 8a, b, 9a, b and 10 are provided as a Source Data file. Source data are provided in this paper.

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

## Acknowledgements

We thank Dr Christian Hacker, Bioimaging Centre at the University of Exeter, for electron microscopy support and Sudheer Gollamudi, Ian Leaves and Rachael Murray for technical support. For the purpose of open access, the author has applied a Creative Commons Attribution (CC BY) licence to any Author Accepted Manuscript version arising from this submission.

## Author contributions

G.S. conceived and coordinated the project, wrote the manuscript, prepared all figures, analysed data and discussed and developed concepts with H.A.B.W. All molecular cloning and *T. reesei* strain generation was performed by S.K. M.S. performed microscopy and analysed data. All authors edited the manuscript and contributed to the writing of the Methods section.

## Competing interests

The authors declare no competing interests.
