## [Transparent Peer Review file · Nature Communications]

Secretion and endocytosis in subapical cells support hyphal tip growth in the fungus *Trichoderma reesei*

Corresponding Author: Professor Gero Steinberg

Version 0:

Reviewer comments:

Reviewer #1

(Remarks to the Author)

In this work, Schuster et al., have done a detailed analysis leading to the important conclusion that subapical hyphal cells support tip growth in the fungus *Trichoderma reesei*. Since fungal hyphal tip growth has always been thought to occur in the tip cell and the subapical hyphal cells were considered to be inactive (or even not alive), this study is indeed ground-breaking and significant. The cell biology work done by the Steinberg group has been regarded highly in the field, and similar to their previous work, the experiments in this current study are well designed and data are also of high quality. I particularly like the experiments in which they used both FRAP (in the presence of a translation inhibitor) and photo-activatable GFP to examine the dynamics of 1,3-beta-glucan synthase. This series of experiments, especially the one using photo-activatable GFP, convincingly show that the cell-wall synthesizing enzyme can be produced in subapical cells and transported to the hyphal tip to support hyphal tip extension. Overall, the work is highly original, results are convincing and well presented. The paper is in general very well written. I strongly support its publication in Nature Communication but would like to provide few points for the authors to consider during revision.

1. Line 56, "references 12,13", regarding secretory vesicles undergoing cytoskeleton-dependent transport, you may consider adding a reference from the Penalva lab who worked on *Aspergillus nidulans* Rab11-marked vesicles, which are considered to be secretory vesicles (for example, Penalva et al., 2017 MBoC).
2. Line 160-161, regarding WB-independent septal pore occlusion, could the Osmani lab paper (Shen et al., 2014, MBoC) be relevant (although this happens upon mitotic entry to keep the pore closed during mitosis)?
3. Line 201-202, "rapid movement" in which direction?
4. Do nuclei divide in the subapical cells? Does *Trichoderma reesei* has septal MTOCs? Answering these questions is not essential to the conclusion of the current paper, but it may be worthwhile to add a few sentences in the discussion.
5. Line 226-227 "current dogma that endocytosis in filamentous fungi is restricted to the hyphal tip cell". You need to add references after the sentence.
6. Line 240-241 "...followed by a switch to random diffusional motility upon release of the endocytic vesicle into the cytoplasm (Fig. 6d)". Has anything similar been observed in other types of cells, for example, budding yeast? If so, it would be better to add a reference.
7. Line 254 "(ER: a targeted codon-optimised enhanced GFP". What is the ER protein in the GFP fusion?
8. Line 390-391 "It is widely accepted that hyphal tip expansion is driven by turgor pressure that generates CS and organelle drift towards the hyphal apex^{40,41}." It would be helpful to discuss where the turgor pressure and cytoplasmic streaming come from so that a reader could get a basic idea without reading the referenced papers.

Reviewer #2

(Remarks to the Author)

Since hyphal growth in filamentous fungi occurs at the hyphal tip, analyses have primarily focused on the tip cell, and the contributions of the second and third subapical cells have rarely been considered. This paper investigates *Trichoderma reesei* and clarifies the roles of the second and third subapical cells behind the tip. Using laser microdissection, the authors lysed cells from the second to the sixth position, demonstrating that the second and third cells are essential for tip growth, while the fourth to sixth cells also play an important role in growth rate. They showed that early endosomes (EE) pass through the septa from the third to the second cell and from the second to the first cell, transporting cargo to the tip. The study also revealed that microtubules penetrate the septa and that secretory vesicles, with their cargo glucan synthases, are transported in a manner similar to EEs, passing from the second to the first cell and then to the tip.

The authors systematically addressed their research questions by leveraging their previous findings in *Ustilago maydis* and insights from the model filamentous fungus *Aspergillus nidulans*, constructing a complete series of fluorescent marker strains for *Trichoderma reesei*. The quality of the live-cell fluorescence imaging is exceptionally high, the dataset is extensive, and the data analysis is appropriate. I generally agree with the conclusions, and by establishing the novel concept of subapical cell contributions to tip growth, this study enhances our understanding of the fundamental principles underlying hyphal growth in filamentous fungi. I support its publication in this journal.

I have some concerns.

First, I would like to know about the timing of septum formation. In other filamentous fungi, I have seen that the 1st and 2nd septa form within a relatively short time frame. Could you provide information on the timing of septum formation in this fungus? Do they form sequentially from the rear in a regular pattern? The reason I am interested in this is that the phenomena demonstrated in this paper might depend on the time elapsed since septum formation. After the tip cell has extended sufficiently and the 1st and 2nd septa have formed, the metabolic activity of the cytoplasm is likely similar to that of the tip cell for a certain period. However, as time passes, metabolic activity (such as protein synthesis and membrane transport) may gradually decline. Reading this paper, I wondered whether septa might exist to accommodate this gradual decline in the activity of the 2nd and 3rd cells. If the septa have just formed, the transport of EE and SV from the subapical cells, as revealed in this study, would not be surprising. How about comparing the elapsed time after the septum is formed with the frequency of membrane transport passing through the septum toward the 1st cell?

While the concept presented in this paper is novel, it does not identify new factors or molecular mechanisms involved in the process, which could be considered a weakness. Nonetheless, I appreciate the valuable and important information provided in Table 1 and Supplementary Table 3.

Regarding endocytosis, previous studies on *A. nidulans* have suggested that endocytosis is more active slightly behind the hyphal tip apex, based on the localization of proteins involved in endocytosis. I understood that, since actin patches are also present in the 2nd and 3rd cells, some level of endocytosis occurs there as well. However, this paper states that previous research suggested that endocytosis does not occur in the 2nd and 3rd cells (L410–411). This seems misleading, as the earlier studies did not explicitly deny endocytosis in the subapical region. I would suggest toning down this statement.

Regarding exocytosis, the paper claims that exocytosis occurs in the region behind the tip because the signal reaches the plasma membrane and disappears there. However, signal disappearance alone is insufficient to confirm exocytosis. Endocytosis was verified by tracking the movement of actin patches, so I suggest confirming exocytosis using the localization of the Exocyst complex, which is involved in this process.

Minor points:

- Fig. 1i: Is the organelle shown a Woronin body?
- Fig. 1j: Later in the paper, EE dynamics are discussed, making this figure clearer. However, at this stage in the manuscript, presenting this figure abruptly makes it difficult to understand.
- Fig. 3: MTOCs at the septum have been reported in *A. nidulans*. Could this be the case in this fungus as well? Additionally, is it known in other filamentous fungi that Lifeact, depending on expression levels, can stabilize actin filaments and create artifacts?
- Fig. 6d, e: The kymograph representation of actin patch dynamics is helpful, but on which line is the change occurring? Also, can you show how the actin patch moves from the plasma membrane toward the cytoplasm?
- L256: ER network and Golgi vesicles are prominent in subapical cells. Does this mean they are also prominent in the 1st apical cell?

Version 1:

Reviewer comments:

Reviewer #1

(Remarks to the Author)

The authors have addressed my comments and have done a nice job revising the paper.

I only have one very minor suggestion:

You may consider adding Zhang et al (from the Fischer lab) 2017 Mol Micro paper when you mentioned *Aspergillus nidulans* septal MTOC.

Reviewer #2

(Remarks to the Author)

The authors have responded to each of the reviewers' requests and questions with great care. Where necessary, they have incorporated additional experiments and new analyses, appropriately improving sections that raised doubts or felt inconsistent. The novelty and significance of this research have been recognized from the outset, and with this revision, it has become more comprehensible and suitable for publication.

Response to Reviewers

Reviewer #1: *In this work, Schuster et al., have done a detailed analysis leading to the important conclusion that subapical hyphal cells support tip growth in the fungus *Trichoderma reesei*. Since fungal hyphal tip growth has always been thought to occur in the tip cell and the subapical hyphal cells were considered to be inactive (or even not alive), this study is indeed ground-breaking and significant. The cell biology work done by the Steinberg group has been regarded highly in the field, and similar to their previous work, the experiments in this current study are well designed and data are also of high quality. I particularly like the experiments in which they used both FRAP (in the presence of a translation inhibitor) and photo-activatable GFP to examine the dynamics of 1,3-beta-glucan synthase. This series of experiments, especially the one using photo-activatable GFP, convincingly show that the cell-wall synthesizing enzyme can be produced in subapical cells and transported to the hyphal tip to support hyphal tip extension. Overall, the work is highly original, results are convincing and well presented. The paper is in general very well written. I strongly support its publication in *Nature Communication* but would like to provide few points for the authors to consider during revision.*

1. Line 56, “references 12,13”, regarding secretory vesicles undergoing cytoskeleton-dependent transport, you may consider adding a reference from the Penalva lab who worked on *Aspergillus nidulans* Rab11-marked vesicles, which are considered to be secretory vesicles (for example, Penalva et al., 2017 MBoC).

Response: We thank the reviewer for this suggestion. Indeed, the mentioned reference is highly suitable and is now included in the text (ln 56) and the reference list (ln 1196-1199).

2. Line 160-161, regarding WB-independent septal pore occlusion, could the Osmani lab paper (Shen et al., 2014, MBoC) be relevant (although this happens upon mitotic entry to keep the pore closed during mitosis)?

Response: Again, a very good suggestion. While Shen et al. focus on occlusion at the onset of mitosis, their results make the point that Woronin body independent sealing of septal pores is possible and may serve specialised, yet crucial functions in the hypha. Thus, we have included the reference in the text (ln 163) and the reference list (ln 1279-1281).

3. Line 201-202, “rapid movement” in which direction?

Response: We admit that we did not think about the direction of SVs in the apical cell, as the current dogma suggests that all SVs move towards the growing tip. However, the reviewer's question prompted us to investigate the transport direction in more detail. Surprisingly, this revealed that SVs move bi-directionally in the entire apical cell. To investigate this further, we performed new laser-

bleaching experiments near the hyphal apex (20-50 um behind the tip). This revealed that long-range retrograde movement of SVs is frequent.

This phenomenon is surprising, and we currently have no explanation why SVs move retrograde. To meet the reviewer's request, we include these data in the manuscript (Abstract: In 37; Results: In 215-220 and In 222-225; New Fig. 4b; new Supplementary Fig. 5a, 5b, 5c; new Supplementary Movie 6). We also analysed retrograde motility of SVs across septa and included these data in Fig. 4f). Regarding discussing this unexpected phenomenon, we decided to not speculate about the significance of this. Instead, we now suggest investigating this phenomenon in a separate study (In. 220). However, we did analyse retrograde cell-to-cell motility across septa, as this communication between the cells is in the focus of our study. This new analysis in the new Fig. 4f.

4. Do nuclei divide in the subapical cells? Does Trichoderma reesei has septal MTOCs? Answering these questions is not essential to the conclusion of the current paper, but it may be worthwhile to add a few sentences in the discussion.

Response: We agree with the reviewer that additional knowledge of the microtubule array in *T. reesei* hyphae is not central to our conclusions. However, as both reviewers asked for more insight into microtubule formation at septa, we addressed both these points. We found that mitotic spindles are formed in all cells that were analysed (Results: 183-187; New Supplementary Fig. 4a). However, not all nuclei in a hyphal cell formed spindles and in the apical cell, long MT bundles remained intact during mitosis. This result is reminiscent of data found in *Aspergillus nidulans* (Horio & Oakley, 2005). We included this reference in the manuscript (In 187 and 1284-1286).

We also found that ~37% of the septa are in contact with a MT end. We depolymerised MTs using benomyl and observed re-growth from septal regions. We found that ~26% of the observed septa (n= 27 experiments; described in Methods, In760-763) show appearance of new MTs in their direct proximity. Thus, we consider it likely that that septal microtubule nucleation sites are present in *T. reesei*. These results, again, correspond with findings in *A. nidulans* (Takeshita & Fischer, 2011). We included this reference in the text (In 193 and 1287-1289) and provide these new data in the results (In 187-194) and in a new Supplementary Figure 4.

5. Line 226-227 "current dogma that endocytosis in filamentous fungi is restricted to the hyphal tip cell". You need to add references after the sentence.

Response: We toned this statement down to meet the request by reviewer 2 (In 244-245). The requested references were added.

6. Line 240-241 "...followed by a switch to random diffusional motility upon release of the endocytic vesicle into the cytoplasm (Fig. 6d)". Has anything similar been observed in other types of cells, for example, budding yeast? If so, it would be better to add a reference.

Response: This behaviour was described in the yeast *Saccharomyces cerevisiae*. We added a note to the text and cited additional relevant literature (ln 256-262 and 1322-1325).

7. Line 254 “(ER: a targeted codon-optimised enhanced GFP”. What is the ER protein in the GFP fusion?

Response: We generated a synthetic reporter, consisting of codon-optimised enhanced GFP that carries a signal sequence from rabbit calreticulin at its N-terminus and an ER retention signal (HDEL) at its C-terminal end. Similar constructs were successfully used by us in *Ustilago maydis* (Wedlich-Soldner et al 2002, Mol Biol Cell) and *Zymoseptoria tritici* (Kilaru et al. 2017, Fungal Genet Biol). We have now added this description in the text (ln. 275-277) and added both references in the legend of Supplementary Figure 1h, where the construct is also described.

8. Line 390-391 “It is widely accepted that hyphal tip expansion is driven by turgor pressure that generates CS and organelle drift towards the hyphal apex^{40,41}.” It would be helpful to discuss where the turgor pressure and cytoplasmic streaming come from so that a reader could get a basic idea without reading the referenced papers.

Response: We agree fully and have now explained the generation of turgor pressure and cytoplasmic streaming in the text (ln. 422-427).

Reviewer 2: *Since hyphal growth in filamentous fungi occurs at the hyphal tip, analyses have primarily focused on the tip cell, and the contributions of the second and third subapical cells have rarely been considered. This paper investigates Trichoderma reesei and clarifies the roles of the second and third subapical cells behind the tip. Using laser microdissection, the authors lysed cells from the second to the sixth position, demonstrating that the second and third cells are essential for tip growth, while the fourth to sixth cells also play an important role in growth rate. They showed that early endosomes (EE) pass through the septa from the third to the second cell and from the second to the first cell, transporting cargo to the tip. The study also revealed that microtubules penetrate the septa and that secretory vesicles, with their cargo glucan synthases, are transported in a manner similar to EEs, passing from the second to the first cell and then to the tip.*

The authors systematically addressed their research questions by leveraging their previous findings in Ustilago maydis and insights from the model filamentous fungus Aspergillus nidulans, constructing a complete series of fluorescent marker strains for Trichoderma reesei. The quality of the live-cell fluorescence imaging is exceptionally high, the dataset is extensive, and the data analysis is appropriate. I generally agree with the conclusions, and by establishing the novel concept of subapical cell contributions to tip growth, this study enhances our understanding of the fundamental principles underlying hyphal growth in filamentous fungi. I support its publication in this journal.

I have some concerns.

1. First, I would like to know about the timing of septum formation. In other filamentous fungi, I have seen that the 1st and 2nd septa form within a relatively short time frame. Could you provide information on the timing of septum formation in this fungus?

Response: If the growing tip cell extends with an average growth speed of 189 μm per hour and the average 2nd cell length is 74 μm (see Table 1 and Supplementary Table 3), the hypha, at average, produces a new septum every 24 minutes.

2. Do they form sequentially from the rear in a regular pattern?

Response: We looked into 18 growing hypha for 1 hour each to investigate if septa are formed in subapical cells. During the 1 hour observation period, the growing tip cell had, at average, produced a new septum every 24 minutes (growth speed 189 μm , septum inserted after \sim 24 min to result in average 2nd cell length of 74 μm ; see Table 1 and Supplementary Table 3). Thus, during these experiments (18 hours), a total of \sim 45 septa were formed in the 1st cell, yet only 1 case of a septum formation in subapical cells was seen. Thus, we consider it likely that septum formation normally occurs at the rear parts of the apical tip cell.

3. The reason I am interested in this is that the phenomena demonstrated in this paper might depend on the time elapsed since septum formation. After the tip cell has extended sufficiently and the 1st and 2nd septa have formed, the metabolic activity of the cytoplasm is likely similar to that of the tip cell for a certain period. However, as time passes, metabolic activity (such as protein synthesis and membrane transport) may gradually decline. Reading this paper, I wondered whether septa might exist to accommodate this gradual decline in the activity of the 2nd and 3rd cells. If the septa have just formed, the transport of EE and SV from the subapical cells, as revealed in this study, would not be surprising.

How about comparing the elapsed time after the septum is formed with the frequency of membrane transport passing through the septum toward the 1st cell?

Response: The main message of this study is that tip growth is supported by subapical cells that undergo endocytic recycling and exocytosis, which enables them to provide growth supplies, including cell wall synthases, to the expanding hyphal apex. The reviewer considers this finding novel and accepts this message.

He/she now asks if the contribution of the individual subunits is depending on the age of the cells. He/she raises the possibility that "young" 2nd cells are still metabolically similar, whereas older cells alter their "metabolic activity of the cytoplasm". He/she suggests that this should be seen in the degree of anterograde membrane trafficking across septa, which, if he/she is right, should gradually decline as cells get older. This is an interesting hypothesis that merits an in-depth answer. We

considered this argument carefully and concluded that aging of the hyphal cells is not likely to explain the degree of trafficking of SVs and EEs from subapical cells. These are our arguments:

When considering the average age of each cell, we note that the subapical cells, analysed here, are indeed maximal 2 hours old (6th cell). This was calculated from the average hyphal elongation rate (~191 $\mu\text{m h}^{-1}$; Supplementary Table 3) and the average length of all cells (1st cell=~189 μm ; all following subapical cells 71-84 μm ; Table 1; Note: We have only considered cells that had a fully formed septum, so the duration of the actual septum formation is not of relevance for this theoretical experiment). As we are only considering fully formed 2nd cell, this compartment could have just been formed by septum formation in the 1st cell, or it could be just before formation of another septum in the first cell, which would turn this 2nd cell into a 3rd cell. The average length of the 2nd cell is 74 μm , which at the average growth speed is formed after ~24 minutes. Consequently, a randomly selected 2nd cell is <1 minute to ~24 minutes old. Applying this calculation to the following subapical cell (using their average length and the average hyphal growth rate), the 3rd cell is ~24- 47 minutes old, the 4th cell is ~47- 71 minutes old, the 5th cell is ~71- 99 minutes old and the 6th cell is ~99- 126 minutes old.

According to the reviewer's argument, the difference in age of the cells should be reflected in the cross-septum motility of SVs and EEs. In other words, if the cross-septum transport is a consequence of the time of maturation and change in metabolic activity, we would expect to see a gradual decline in this anterograde cell-to-cell transport. However, we do not see this for SV motility (Fig. 4f). In fact, anterograde SV motility through all 5 septa is not statistically different (one-way ANOVA test, 2-tailed P value of 0.0586), but even increased in Septum 4.

We do, however, realise that EE motility differs amongst the 5 septa (Fig. 5f; one-way ANOVA test, 2-tailed P value of 0.0002), with more anterograde transport through septum 1 and 2 than through septum 3- 5 (Fig. 5f). Nevertheless, transport through septum 3- 5 still happens and is not significantly different (one-way ANOVA test, 2-tailed P value of 0.9823), although these cells differ in their age. Thus, we conclude that the first 2 subapical cells provide more EEs to the apical cell, but this appears not be due to an unspecific aging process. We have added this a note to the Discussion to highlight this finding (In 404-406).

Another line of argument against increasingly inactive subapical cells comes from a study that used Raman microspectroscopy in *A. nidulans* hyphae (Yasuda et al. 2021, Sci. Rep. 11:1279). Here protein synthesis in the apex of hyphae was investigated as a marker of metabolic activity. Surprisingly large differences were found in the confined space of the hyphal tip (a drop by ~1.8-times over a distance of <5 μm). Moreover, the increased activity was related to endocytic activity at the hyphal apex. We report here that endocytic activity within the 1st cell drops by ~2-times over the length of the entire 1st cell and stays stable at this level in all subapical cells (Fig. 6f; one-way ANOVA test of actin patch numbers at the rear end of the 1st cell and the following subapical 2nd to 6th cell reveals no significant difference; two-tailed P-value of 0.3949). Obviously, actin patch formation/endocytic activity is not a direct measure for metabolic activity. However, subapical cells are, regarding endocytosis, as active in as they are in the rear parts of the tip cell. This makes a decline in metabolic activity with aging of the cells unlikely.

Finally, we wish to consider distribution of the endoplasmic reticulum and the Golgi apparatus. Our quantitative analysis of GFP-reporter proteins shows a decline of both organelles in the 2nd cell. This further declines in the 3rd cell, yet no further drop was found in the 4th-6th cell (Fig. 7a, 7b). Again, these results support the notion that the 2nd cell is more active, yet all other subapical cells are similar in their metabolic activity/protein synthesis capacity, despite being of different age.

In summary, we conclude that the argument by the reviewer might well apply to the 2nd cell, as these cells can be as young as a few minutes. Indeed, the 2nd cell contains more Golgi cisternae and more ER than the other subapical cells. Moreover, together with the 3rd cell, it provides cytoplasmic streaming in the hypha and delivers more EEs to the tip cell. This additional, previously not noticed, contribution to tip growth puts the 2nd cell even more firmly in the "Core Growth Unit", defined in this paper (see Fig. 9e).

Response: We are very grateful to the reviewer for raising this point. It prompted us to look at the data from a different angle. We highlighted our conclusion about the increased importance of the 2nd cell now more explicitly in the Discussion (ln 404-406).

4. While the concept presented in this paper is novel, it does not identify new factors or molecular mechanisms involved in the process, which could be considered a weakness. Nonetheless, I appreciate the valuable and important information provided in Table 1 and Supplementary Table 3.

Response: Indeed, we did not intend to identify any new molecule. We were addressing the much more global concept of hyphal growth and consider the conceptual change that this paper brings very important.

5. Regarding endocytosis, previous studies on A. nidulans have suggested that endocytosis is more active slightly behind the hyphal tip apex, based on the localization of proteins involved in endocytosis. I understood that, since actin patches are also present in the 2nd and 3rd cells, some level of endocytosis occurs there as well. However, this paper states that previous research suggested that endocytosis does not occur in the 2nd and 3rd cells (L410–411). This seems misleading, as the earlier studies did not explicitly deny endocytosis in the subapical region. I would suggest toning down this statement.

Response: We need to highlight here that the phrase "some actin patches occur in the 2nd and 3rd cell" is not accurate. Firstly, we report that actin patches are found in all 5 subapical cells investigated. Secondly, our quantitative analysis of actin patch numbers (as an indicator of endocytic activity) shows that each subapical cell shows ~25-30% of the activity of the tip cell. This number takes into account that subapical cells are much shorter than the tip cell. When we directly compare areas of 1 μm^2 , the activity in subapical cells reaches almost ~40-50% of that in hyphal tips. Thus, we consider the subapical endocytic activity very significant. This new calculation was added to Table 1 and is also mentioned in the text (ln 445-450).

However, we agree that the papers cited did not explicitly deny endocytosis in subapical cells of the hypha (they just did not consider these cells). We therefore toned down our statement as requested (In 244-245 and 448-450).

6. Regarding exocytosis, the paper claims that exocytosis occurs in the region behind the tip because the signal reaches the plasma membrane and disappears there. However, signal disappearance alone is insufficient to confirm exocytosis. Endocytosis was verified by tracking the movement of actin patches, so I suggest confirming exocytosis using the localization of the Exocyst complex, which is involved in this process.

Response: We thank the reviewer for this valuable comment. Indeed, the exocyst tethers exocytic vesicles to the plasma membrane before the SNARE complex is formed. The exocyst protein Exo70p is located at sites of exocytosis at the plasma membrane in yeast (Boyd et al. 2004, J Cell Biol). We introduced the gene encoding codon-optimised enhanced GFP to the endogenous *exo70* homologue gene in *T. reesei* and visualised the native levels of the fusion protein in hyphae. We found the exocyst subunit in the plasma membrane of subapical cells. This result adds more support to our conclusion that subapical exocytosis exists in subapical hyphal cells of *T. reesei*.

We show the localisation of TreGFP-Exo70 in the plasma membrane in Fig. 7h, describe the data in the main text (Results: 303-309; Figure legend 7h: In 1085-1088; Methods: In 528-529, In 607-608 and In 648-650; ; Reference 61, In 1338-1340) and in Supplementary Information: Supp. Table 1, Supp. Table 2, Supp. Table 4, Supp. Table 5, Supp. Figure 1l, Supp. Figure legend 1l; new reference 13; Supp. Plasmid Cloning: page 18/19)

Minor points:

7. Fig. 1i: Is the organelle shown a Woronin body?

Response: The dark organelle shown in Fig. 1i and Supplementary Movie 3 is able to seal the septum despite high internal pressure (illustrated by the bending of the septum after "plugging" the pore; Supplementary Movie 3). To our knowledge, only Woronin bodies are rigid enough to withstand the internal pressure, which is due to their crystallin core of Hex1-protein. Thus, we consider it most likely that the dark organelle shown in this Figure and in the Supplementary Movie 3 is, indeed, a Woronin body.

8. Fig. 1j: Later in the paper, EE dynamics are discussed, making this figure clearer. However, at this stage in the manuscript, presenting this figure abruptly makes it difficult to understand.

Response: In Fig. 1j, EEs are only used as a fluorescent marker of cytoplasmic bulk flow. However, we appreciate the concerns of the reviewer and tried to explain this experiment and the use of EEs better (In 138-142).

9. Fig. 3: MTOCs at the septum have been reported in *A. nidulans*. Could this be the case in this fungus as well?

Response: This point was also raised by Reviewer 1 (point 4.), and as outline above we acknowledged this interest, although the outcome of this analysis is of limited importance for the main conclusions of this study.

We analysed Z-axis image stacks of >60 cells and asked if MTs end at a septum. We found that ~37% of the septa are in contact with a MT end. Next, we depolymerised MTs using benomyl and observed re-growth from septal regions in 26% of all septa observed. Thus, we consider it likely that that septal microtubule nucleation sites are present in *T. reesei*. These results, again, correspond with findings in *A. nidulans* (Takeshita & Fischer, 2011). We included this reference in the text (ln 193 and 1287-1289) and provide these new data in the results (ln 187-194) and in a new Supplementary Figure 4c.

10. Additionally, is it known in other filamentous fungi that Lifeact, depending on expression levels, can stabilize actin filaments and create artifacts?

Response: Based on numerous publications, Lifeact is the best F-actin probe that is available. However, we do, indeed, find that expression of Lifeact can result in abnormal hyphal morphology in *T. reesei*. We therefore investigated only cells that (i) showed a normal morphology and (ii) were still growing while observing F-actin. In doing so, we think we have minimised the risk of artifacts.

11. Fig. 6d, e: The kymograph representation of actin patch dynamics is helpful, but on which line is the change occurring?

Response: We are not clear what the reviewer is asking here. These kymographs were derived from image sequences of top-views. We do not know what lines the reviewer is referring to. However, we added an arrow to indicate the transition point from stationary to random motility.

12. Also, can you show how the actin patch moves from the plasma membrane toward the cytoplasm?

Response: We have included kymograph that shows a side-view of an dynamic actin patch in the new Fig. 4c.

13. L256: ER network and Golgi vesicles are prominent in subapical cells. Does this mean they are also prominent in the 1st apical cell?

Response: Yes, they are, which is shown in Fig. 7a and 7b. We explained this statement better in the revised version of our manuscript (ln 278-280).

Response to Reviewers

Reviewer #1: *The authors have addressed my comments and have done a nice job revising the paper. I only have one very minor suggestion:*

1. *You may consider adding Zhang et al (from the Fischer lab) 2017 Mol Micro paper when you mentioned Aspergillus nidulans septal MTOC.*

Response: We thank the reviewer for this suggestion and included the reference in the text (In 192).

Reviewer 2: *The authors have responded to each of the reviewers' requests and questions with great care. Where necessary, they have incorporated additional experiments and new analyses, appropriately improving sections that raised doubts or felt inconsistent. The novelty and significance of this research have been recognized from the outset, and with this revision, it has become more comprehensible and suitable for publication.*

Response: We thank the reviewer for the positive comments